

# Massive holographic QFTs in de Sitter

**José Manuel Penín[1,2⋆], Kostas Skenderis[2†] and Benjamin Withers[2‡]**

**1** Department of Physics and Helsinki Institute of Physics, P.O.Box 64,
FIN-00014 University of Helsinki, Finland
**2** Mathematical Sciences and STAG Research Centre, University of Southampton,
Highfield, Southampton SO17 1BJ, UK

⋆ jmanpen@gmail.com , † k.skenderis@soton.ac.uk , ‡ b.s.withers@soton.ac.uk

## Abstract

We study strongly coupled mass-deformed-CFT on a fixed de Sitter spacetime in three dimensions via holography. We elucidate the global causal structure of the four-dimensional spacetime dual to the de Sitter invariant vacuum state. The conformal boundaries of de Sitter appear as spacelike defects sourced by the mass deformation, which extend into the bulk as curvature singularities in AdS. We compute all one- and two-point functions of the deformed-CFT stress tensor and a scalar operator order-by-order in the mass deformation for a simple holographic model. These correlation functions admit a spectral representation as a sum of simple poles corresponding to normalisable modes in the bulk.

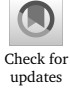

## 1   Introduction

De Sitter spacetime is both physically relevant and mathematically significant. Its physical relevance is its description of a universe expanding at an accelerating rate, applicable at early times in inflationary cosmology, and at late times as we enter an era dominated by a cosmological constant. It is also of mathematical significance as one of only three maximally symmetric Lorentzian geometries. For these reasons it is important to develop a good understanding of quantum field theories (QFT) on de Sitter backgrounds.

Weakly coupled QFTs on fixed de Sitter is a well-studied and rich field of research. Light scalars in de Sitter backgrounds ($|m| \ll H$) exhibit infrared divergences if interactions are treated as perturbatively small. In Starobinksky's stochastic approach [1–3], this is addressed by splitting fields into long- and short-wavelength parts compared to the Hubble scale. The long wavelength parts then evolve classically and stochastically according to a Langevin equation with a white noise term arising from short wavelength modes. Starobinsky's approach, and generalisations of it, have been the focus of many subsequent studies [4–22]. For reviews on infrared issues in de Sitter see [23, 24].

In this work we adopt a fresh perspective and compute QFT correlation functions on de Sitter directly at strong coupling. This sidesteps issues encountered by working with perturbatively small coupling constants, and at the same time provides a complementary insight into general properties of correlation functions on de Sitter backgrounds at any value of the coupling. Strong coupling is made accessible through holographic duality, where we may set the QFT to live on a fixed de Sitter metric with Hubble parameter $H$,

$$ds_{dS_3}^2 = -dt^2 + e^{2Ht}d\vec{y}^2 = \frac{-d\eta^2 + d\vec{y}^2}{H^2\eta^2}, \tag{1}$$

where the conformal time $\eta = -H^{-1}e^{-Ht}$. Conformal field theories (CFT) provide the best understood examples of holographic duality, however since de Sitter spacetime (1) is in the same conformal class as Minkowski they do not capture any dynamics for which the curved de Sitter background plays an important role.[1] This motivates the study of non-conformal field theories.

---

[1]Up to the role played by possible conformal anomalies.

Therefore here we turn our attention to non-conformal holographic QFTs. This is achieved by introducing a mass as a deformation parameter of a CFT,

$$S = S_{\text{CFT}} + \int d^d x\, m(x) O(x). \tag{2}$$

Since $m$ breaks conformal symmetry there is no longer a relation to vacuum QFT on Minkowski spacetime under a Weyl transform, instead, Weyl transformations reveal an equivalence to studying QFT on Minkowski spacetime in the presence of a spacelike defect. The defect is located at the future conformal boundary of de Sitter, $\eta \to 0$. The vacuum state in de Sitter is characterised by constant energy density despite the accelerated expansion; one interpretation of this is a balance between particle production and its dilution due to the expansion of the universe. The bulk global structure of the vacuum is given by domain wall solutions in a dS-invariant foliation of the bulk, where the leaves of the foliation are appropriately marshalled by the defect. The geometries are akin to a Janus solution, but where the defect is spacelike and separates two copies of de Sitter.

Such states have been considered in a series of previous works [25–29] and also more recently in [30–34]. In these papers they are shown to exhibit properties commensurate with dynamical attractors; homogeneous deviations from vacuum relax exponentially fast in time (power law in the scale factor) exhibiting features which elicit comparison to hydrodynamic equilibrium and quasinormal modes. It is interesting to ask how far the analogy to hydrodynamic equilibrium and quasinormal modes goes, especially since the state is not an equilibrium state. To this end, we compute two-point functions of currents as the ultimate arbiter of relaxation towards any given state. We do this for scalar, vector and tensor channels and at finite spatial momentum. We find no further similarities to hydrodynamics; the underlying reason is the high amount of residual isometries. Since the domain walls are dS invariant there can be no meaningful dispersion at finite momentum as would be the case in hydrodynamics, only modes organised into de Sitter eigenfunctions.

For completeness we would like to point out several other holographic treatments of cosmological spacetimes in the literature. In this work we place importance on breaking away from conformal theories and we do so with a mass term. Another way to break conformality is to work with confining spacetimes, for instance by spatially compactifying the spacetime to $dS_d \times S_1$ [35–38]. In a distinct paradigm, one may consider working with the quantum theory of gravity with $dS_{d+1}$ future asymptotics working through a Euclidean QFT dual [39–41]. Or holographic treatments of the static patch of $dS_d$, [42]. Within fluid-gravity, a cosmological expansion can be introduced provided the scale factor is treated within a derivative expansion [43].

The paper is organised as follows. In section 2 we present the holographic model, the geometry corresponding to the vacuum state in the presence of a mass, including its global causal structure as it embeds into $AdS_4$. In section 3 we compute the one-point functions for this state. In section 4 we perform a gauge-invariant decomposition of fluctuations organised by dS isometries and compute two-point functions and normalisable mode spectra. In section 5 we present a novel Bessel function basis for expressing CFT two-point functions, a by-product of our analysis. In section 6 we present a free fermion calculation and compare with our strong-coupling holographic results. We finish with a discussion of results and future directions in section 7.

## 2 Bulk geometry for the massive dS$_3$ vacuum

Throughout this work, where a specific holographic model is required we adopt the following bulk action,

$$S = \frac{1}{2\kappa^2} \int_{\mathcal{M}_4} \sqrt{-G} d^4 x \left( R + 6 - \frac{1}{2} (\partial \phi)^2 + \phi^2 \right), \tag{3}$$

where the AdS$_4$ radius $L = 1$ and the bulk scalar mass term (distinct from the mass deformation we consider) corresponds to a $\Delta = 2$ scalar operator in the dual field theory. To construct the corresponding bulk solutions we adopt the following domain wall ansatz,

$$ds^2 = G_{ab}^{\text{DW}} dX^a dX^b = dz^2 - P(z) ds_{dS_3}^2, \qquad \phi = \bar{\phi}(z), \tag{4}$$

where $ds_{dS_3}^2$ is the dS$_3$ metric given in (1), resulting in the following equations of motion,

$$\bar{\phi}'' + \frac{3P'}{2P} \bar{\phi}' + 2\bar{\phi} = 0, \tag{5}$$

$$6H^2 + \frac{3(P'^2)}{2P} - \frac{1}{2} P \left( 12 + 2\bar{\phi}^2 + (\bar{\phi}')^2 \right) = 0. \tag{6}$$

See [44] for a general treatment of such domain walls. In accordance with the preceding discussion, we wish to impose boundary conditions at the conformal boundary (here taken to be $z \to -\infty$) which implements the mass-deformation of the theory. In detail, near the boundary as $z \to -\infty$ we have the following expansion

$$P = e^{-2z} \left( P_{(0)} + P_{(2)} e^{2z} + P_{(3)} e^{3z} + \dots \right), \tag{7}$$

$$\bar{\phi} = \bar{\phi}_{(1)} e^z + \bar{\phi}_{(2)} e^{2z} + \dots, \tag{8}$$

and we impose that the boundary metric is given by $ds_{dS_3}^2$ and the scalar sources a mass deformation, *viz.*

$$P_{(0)} = -1, \qquad \bar{\phi}_{(1)} = m. \tag{9}$$

The bulk solutions are constructed by imposing regularity for a certain region of the spacetime. This is best understood by referring to the global structure of the spacetime which we will elucidate momentarily. The final outcome of this analysis is the following bulk solution, which can be constructed analytically as a perturbative expansion in $m$,

$$P = -e^{-2z} \left( 1 - \frac{H^2}{4} e^{2z} \right)^2 - \frac{(-144 + 112He^z - 32H^2e^{2z} + 4H^3e^{3z} + H^4e^{4z})}{1152 \left( 1 + \frac{H}{2} e^z \right)^2} m^2 + O(m)^4,$$

$$\bar{\phi} = \frac{e^z}{\left( 1 + \frac{H}{2} e^z \right)^2} m - \frac{e^{2z}(40 + 12He^z + 14H^2e^{2z} + H^3e^{3z})}{576H \left( 1 + \frac{H}{2} e^z \right)^6} m^3 + O(m)^5. \tag{10}$$

This corresponds to the solutions detailed in ingoing coordinates in [27]. In appendix A we present the coordinate transformation between the ingoing coordinate system and the domain wall coordinates used here. The corresponding one point functions for these geometries are presented in section 3 after performing appropriate renormalisation.

### 2.1 Global structure

The above solutions are constructed in coordinates such that the boundary covers the inflationary patch of dS$_3$, and the portion of the bulk region covered depends on whether domain wall (4) or ingoing (110) coordinates are used. The goal of this section is to elucidate the global

structure by appealing to the complete spacetime. We start this discussion on the boundary where we identify the requisite Weyl transformation, and then turning to the bulk coordinate transformations which implement it. Our approach is to start at $m = 0$ where the leading scalar behaviour is a probe, remarkably this leads to the identification of a convenient ansatz which elucidates the global structure of the geometry at finite $m$.

### 2.1.1 Weyl

On the boundary, we first perform a Weyl transformation which maps from the inflationary patch of $dS_3$ to Minkowski spacetime. This is straightforward once considering conformal time parameter $\eta = -H^{-1}e^{-Ht}$ for which the boundary metric (1) becomes,

$$\frac{-d\eta^2 + d\vec{y}^2}{H^2\eta^2} \xrightarrow{\text{Weyl}} -d\eta^2 + d\vec{y}^2 \,. \tag{11}$$

Under this Weyl transformation, the deformation (9) transforms as follows,

$$\bar{\phi}_{(1)} = m \xrightarrow{\text{Weyl}} \frac{m}{-H\eta} \,, \tag{12}$$

and hence the future conformal boundary of $dS_3$ is described by a singular spacelike source function in $\mathbb{R}^{1,2}$ resembling a defect. This singularity extends into the bulk as we shall now show.

### 2.1.2 Global bulk, $m = 0$

At $m = 0$ where the bulk geometry is identically $AdS_4$, given by the first term for $P(z)$ in (10). Consider the following bulk coordinate transform which implements the Weyl transform (11),

$$z = \log\left(-\frac{r}{H\tau}\frac{2\tau^2 - 2\sqrt{\tau^4 - r^2\tau^2}}{r^2}\right), \qquad \eta = \tau\frac{\tau^2 - r^2 - \sqrt{\tau^4 - r^2\tau^2}}{\tau^2 - \sqrt{\tau^4 - r^2\tau^2}} \,. \tag{13}$$

In addition this transform obeys the following features: it preserves the time coordinate on the boundary, i.e. $\lim_{r\to 0}\eta = \tau$, and satisfies $z \to -\infty$ as $r \to 0$ when $\eta < 0$ in accordance with the boundary limit. Under this transformation the leading bulk line element takes on standard Poincaré form together with the leading bulk scalar contribution (10),

$$ds^2 = r^{-2}(dr^2 - d\tau^2 + d\vec{y}^2) + O(m)^2, \qquad \bar{\phi} = \frac{r}{r-\tau}\frac{m}{H} + O(m^3). \tag{14}$$

Hence the singularity on the boundary at $\tau = 0$ extends into the bulk along the null surface $\tau = r$, to leading order in $m$. Whilst informative, the Poincaré patch is still only a portion of the full spacetime. The next step along the path to the global structure is mapping from the plane to the Lorentzian cylinder, $R_t \times S^2$, such that the bulk solution at $m = 0$ is written in global $AdS_4$ coordinates. First switching to polar coordinates on the boundary, $d\vec{y}^2 = dR^2 + R^2 d\Phi^2$, then let

$$r = \frac{\sin\bar{r}}{\cos T + \cos\theta\cos\bar{r}}\,, \qquad \tau = \frac{\sin T}{\cos T + \cos\theta\cos\bar{r}}\,, \qquad R = \frac{\sin\theta\cos\bar{r}}{\cos T + \cos\theta\cos\bar{r}} \tag{15}$$

the bulk metric and scalar become

$$ds^2 = \csc^2\bar{r}\left(-dT^2 + d\bar{r}^2 + \cos^2\bar{r}\,d\Omega_2^2\right) + O(m)^2, \qquad \bar{\phi} = \frac{\sin\bar{r}}{\sin\bar{r} - \sin T}\frac{m}{H} + O(m)^3, \tag{16}$$

where $d\Omega_2^2 = d\theta^2 + \sin^2\theta\,d\phi^2$. Here the boundary is at $\bar{r} = 0$ and the origin is at $\bar{r} = \pi/2$. The profile of $\bar{\phi}$ shows that the singularity (within the domain $T \in [-\pi, \pi]$) is located along the null surfaces $T = \bar{r}$ and $T = \pi - \bar{r}$, to leading order in $m$.

### 2.1.3 Global bulk, $m \neq 0$

Remarkably, the chain of coordinate transformations which implements the boundary Weyl transformation to the Lorentzian cylinder, discussed above at $m = 0$, generalises straightforwardly to arbitrary $m$. The final general vacuum solution takes the form of AdS$_4$ written in global coordinates up to an overall conformal factor, $\Omega$, as follows:

$$ds^2 = \Omega(Y)^2 \csc^2 \bar{r} \left( -dT^2 + d\bar{r}^2 + \cos^2 \bar{r}\, d\Omega_2^2 \right), \tag{17}$$

$$\bar{\phi} = F(Y), \tag{18}$$

where we have introduced

$$Y \equiv \frac{\sin T}{\sin \bar{r}}. \tag{19}$$

As such, identification of the causal structure of the solutions is now straightforward as the conformal diagram is identical to AdS$_4$, up to the locations of bulk singularities. The resulting equations of motion are ODEs in the variable $Y$ alone, a manifestation of the underlying dS$_3$ isometries of (17). In particular, the isometries of (17) are given by the dS$_3$ isometry group, realised by the following 6 Killing vectors,

$$\xi_D = \sin T \cos \bar{r} \cos \theta\, \partial_T + \cos T \sin \bar{r} \cos \theta\, \partial_{\bar{r}} + \cos T \sec \bar{r} \sin \theta\, \partial_\theta, \tag{20}$$

$$\begin{aligned} \xi_{P_1} = & -\sin T \cos \bar{r} \sin \theta \cos \Phi\, \partial_T - \cos T \sin \bar{r} \sin \theta \cos \Phi\, \partial_{\bar{r}} \\ & + (1 + \cos T \sec \bar{r} \cos \theta) \cos \Phi\, \partial_\theta - (\cot \theta + \cos T \csc \theta \sec \bar{r}) \sin \Phi\, \partial_\Phi, \end{aligned} \tag{21}$$

$$\begin{aligned} \xi_{P_2} = & -\sin T \cos \bar{r} \sin \theta \sin \Phi\, \partial_T - \cos T \sin \bar{r} \sin \theta \sin \Phi\, \partial_{\bar{r}} \\ & + (1 + \cos T \sec \bar{r} \cos \theta) \sin \Phi\, \partial_\theta + (\cot \theta + \cos T \csc \theta \sec \bar{r}) \cos \Phi\, \partial_\Phi, \end{aligned} \tag{22}$$

$$\begin{aligned} \xi_{K_1} = & \sin T \cos \bar{r} \sin \theta \cos \phi\, \partial_T + \cos T \sin \bar{r} \sin \theta \cos \Phi\, \partial_{\bar{r}} \\ & + (1 - \cos T \sec \bar{r} \cos \theta) \cos \Phi\, \partial_\theta - (\cot \theta - \cos T \sec \bar{r} \csc \theta) \sin \Phi\, \partial_\Phi, \end{aligned} \tag{23}$$

$$\begin{aligned} \xi_{K_2} = & \sin T \cos \bar{r} \sin \theta \sin \phi\, \partial_T + \cos T \sin \bar{r} \sin \theta \sin \Phi\, \partial_{\bar{r}} \\ & + (1 - \cos T \sec \bar{r} \cos \theta) \sin \Phi\, \partial_\theta + (\cot \theta - \cos T \sec \bar{r} \csc \theta) \cos \Phi\, \partial_\Phi, \end{aligned} \tag{24}$$

$$\xi_M = \partial_\Phi, \tag{25}$$

where $\theta, \Phi$ are the polar and azimuthal angle on the S$^2$, and the vectors have been labelled in a way which indicate their role in the Euclidean global conformal algebra in two dimensions. Each of these leave $Y$ invariant. The boundary of this spacetime is the Einstein static universe $R \times S^2$, and the dS$_3$ region of interest is conformal to a piece of this, as illustrated in figure 1.

Some comments on the privileged coordinate $Y$ are now in order. The AdS boundary to the past of $\eta = 0$ is reached by $Y \to -\infty$ (i.e. the blue square region in figure 1), and the AdS boundary to the future of $\eta = 0$ is reached by $Y \to +\infty$ (i.e. the white square region in figure 1), with a fixed point corresponding to the location of the defect, i.e. the singularity in the source function. The values $Y = \pm 1$ correspond to radial null lines departing from and arriving at the defect, respectively. This is illustrated in figure 2 where contours of $Y$ are shown on the conformal diagram for the solution, which is identical to the conformal diagram for global AdS up to the inclusion of the new singularities in red.

Starting with this new ansatz, (17), (18) we can revisit the perturbative analysis. In particular, the solution at leading order in $m$ is given by,

$$\Omega^2(Y) = 1 + O(m)^2, \tag{26}$$

$$F(Y) = \frac{Y + c_F}{(Y+1)(1-Y)} \frac{m}{H} + O(m)^3. \tag{27}$$

This leaves a clear choice to either eliminate the incoming singularity at $Y = -1$ or the outgoing singularity at $Y = 1$. Eliminating the incoming singularity (choosing $c_F = 1$) and continuing

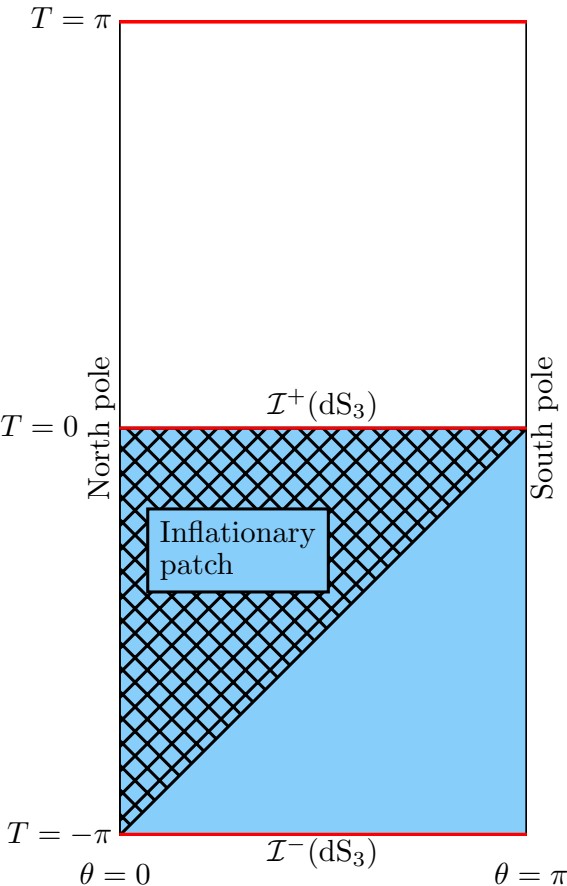

Figure 1: $dS_3$ is conformal to a portion of $R \times S^2$, here shown as a blue region (the azimuthal angle $\Phi$ is suppressed). The spacelike conformal boundaries of dS are shown in red, and correspond to the singular loci of a mass deformation of a CFT living in $dS_3$. The infilling bulk geometry is conformal to $AdS_4$, given by (17) when foliated by constant $Y$ surfaces. The blue square region corresponds to the part of the conformal boundary reached when $Y \to -\infty$ and the white square region to $Y \to +\infty$.

to higher orders in $m$ gives,

$$\Omega^2(Y) = 1 - \frac{1}{12(Y-1)^2}\frac{m^2}{H^2} - \frac{5}{432(Y-1)^3}\frac{m^4}{H^4} + O(m)^6, \tag{28}$$

$$F(Y) = \frac{1}{1-Y}\frac{m}{H} + \frac{3-5Y}{72(Y-1)^3}\frac{m^3}{H^3} + \frac{-175+619Y-645Y^2+129Y^3}{51840(Y-1)^5}\frac{m^5}{H^5} + O(m)^7, \tag{29}$$

which matches the geometry (10) up to a coordinate transformation.

We now turn our attention to the location of the singularity at finite $m$. The perturbative expressions for the scalar $F(Y)$ make this identification subtle, since each order in $m$ comes with an additional factor of $(1-Y)^{-1}$ which naively appears to invalidate the perturbative expansion there. Non-perturbatively, we find that this coincides with the existence a logarithmic singularity by constructing a solution in the neighbourhood of a fiducial singular point $Y_*$,

$$F = \pm\sqrt{3}\log(Y-Y_*) + F_0 + F_1(Y-Y_*) + \dots, \qquad \Omega^2 = h_0(Y-Y_*) + \dots. \tag{30}$$

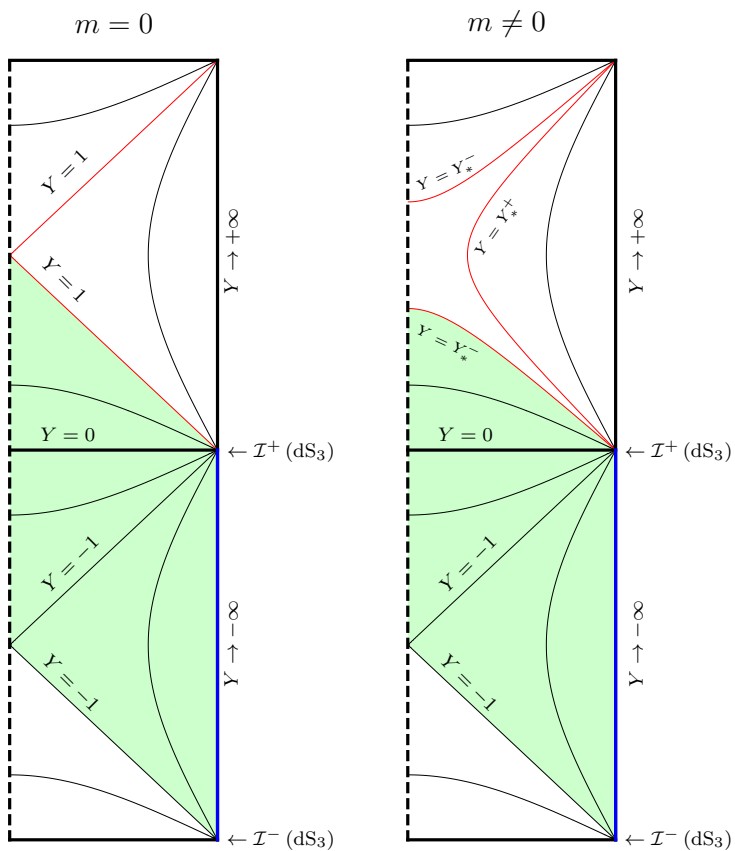

Figure 2: Conformal diagrams corresponding to the global extension (17), showing the effect of introducing the mass deformation to the field theory on $dS_3$, $m$. Each point is an $S^2$ which shrinks to zero size at the origin of coordinates indicated by the dashed line, and the remaining lines are level sets of $Y = \frac{\sin T}{\sin \bar{r}}$. $Y \to -\infty$ corresponds to a portion of the AdS boundary $R \times S^2$ before the defect, i.e. the blue line in this figure, corresponding to the blue square region shown in figure 1. $Y \to +\infty$ corresponds to the complement of the blue region on the boundary, i.e. the white square region shown in figure 1. The green shaded region shows the development of data prescribed in the blue $dS_3$ region at the boundary together with suitable choice of vacuum along the past portion of null surface at $Y = -1$. **Left panel:** At $m = 0$ the bulk is exactly $AdS_4$ and the probe scalar singularity is null, as shown by the red lines along $Y = 1$ in the left panel. **Right panel:** With $m \neq 0$ the bulk is conformal to $AdS_4$ and the singularity is split into timelike and spacelike components at $Y = Y_*^\pm$ in (31), shown in red.

This behaviour has been confirmed by direct numerical construction in the $Y$-variable. With the singularity characterised, $Y_*$ can be computed perturbatively in $m$ by matching the functional form of $(F')^{-1}$ in the vicinity of the singularity, finding two such solutions $Y_* = Y_*^\pm$,

$$Y_*^\pm = 1 \pm \frac{1}{2\sqrt{3}} \frac{m}{H} + O(m)^2. \tag{31}$$

Thus, $m$ causes the singularity to split into a timelike and spacelike component, as illustrated in the right panel of figure 2. Additionally, at this point $\Omega$ vanishes order by order in $m$. Going beyond perturbation theory, we can numerically construct solutions by solving the ODEs in $Y$

at finite $m$. This is achieved by a shooting method, between an expansion near the boundary at $Y = -\infty$ where we impose the source amplitude $m$, and an expansion about $Y = -1$ constructed to be manifestly regular there. By counting data appearing in these expansions and comparing to the order of the ODEs one concludes that there is indeed a one-parameter family of solutions which can be labelled by $m$. Once such a solution is constructed, it can be further integrated from $Y = -1$ towards $Y = 1$, reading off the location of the singularity encountered along the way at $Y_*^-$. The results of this exercise are shown in figure 3 and show agreement with the perturbative calculation (31) for small $m$.

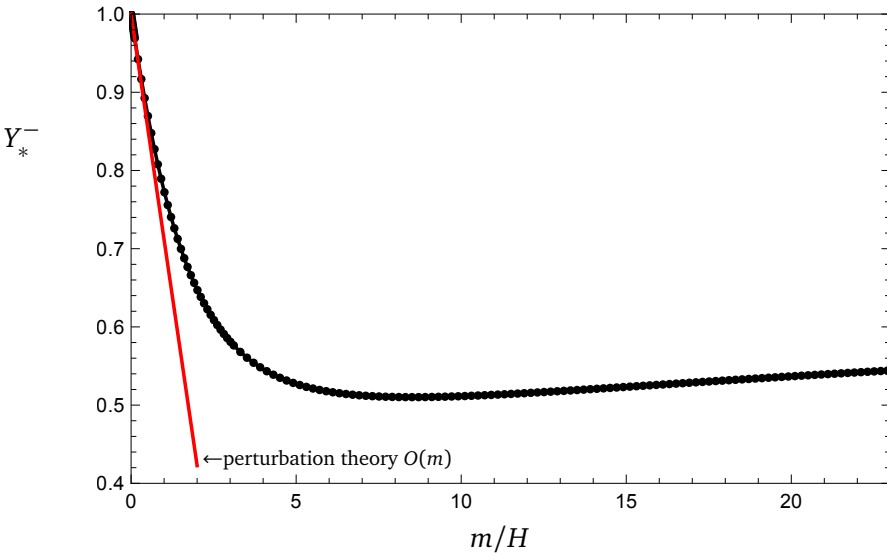

Figure 3: The location of the spacelike singularity $Y_*^-$ as a function of the mass deformation parameter $m$. The red line shows the leading slope predicted by perturbation theory, i.e. the negative branch of (31).

## 3 One-point functions

The only two scales available are the mass deformation $m$ and the Hubble constant $H$. Furthermore, the bulk geometry is manifestly dS$_3$ invariant. The one point functions therefore must take the form,

$$\langle O \rangle_0 = \frac{H^2}{2\kappa^2} \mathcal{F}\left(\frac{m}{H}\right), \tag{32}$$

$$\langle T_{\mu\nu} \rangle_0 = -\frac{H^3}{2\kappa^2} \frac{m}{3H} \mathcal{F}\left(\frac{m}{H}\right) g_{\mu\nu}^{dS}, \tag{33}$$

where the relationship between the two expressions is in accordance with the trace Ward identity, (126), ensuring that the same function $\mathcal{F}$ appears in both. The background geometries are given in the domain wall form (4), and to convert these to Fefferman-Graham gauge is straightforward, requiring only a change of the bulk radial variable,

$$z = \frac{1}{2} \log \rho. \tag{34}$$

With this mapping complete we can simply read off the one-point functions from the near boundary data according to (132) and (133). See appendix B for details of the holographic

renormalisation procedure. In a neighbourhood around $m = 0$ utilising the perturbative solutions we find,

$$\mathcal{F} = -\frac{m}{H} - \frac{5}{72}\frac{m^3}{H^3} + \frac{43}{17280}\frac{m^5}{H^5} + O(m)^7, \tag{35}$$

in agreement with the results of [27]. As $m/H \to \infty$ the scale $m$ dominates and is the only scale that enters the expressions (32) and (33). This is reflected in the asymptotic behaviour of $\mathcal{F}$ at large argument, given by

$$\mathcal{F} = \mathcal{F}_{\text{asy}}\frac{m^2}{H^2}, \tag{36}$$

for some constant $\mathcal{F}_{\text{asy}}$. We can determine this constant by appealing to the bulk equations of motion, and appropriately scaling the bulk radial coordinate by $m$ and taking the large $m$ limit. This leads to an exact set of equations governing the large $m$ behaviour.[2] Such a numerical analysis gives the approximate value $\mathcal{F}_{\text{asy}} \simeq -0.37$. These two asymptotic regions are connected numerically for all $m$ as shown in figure 4.

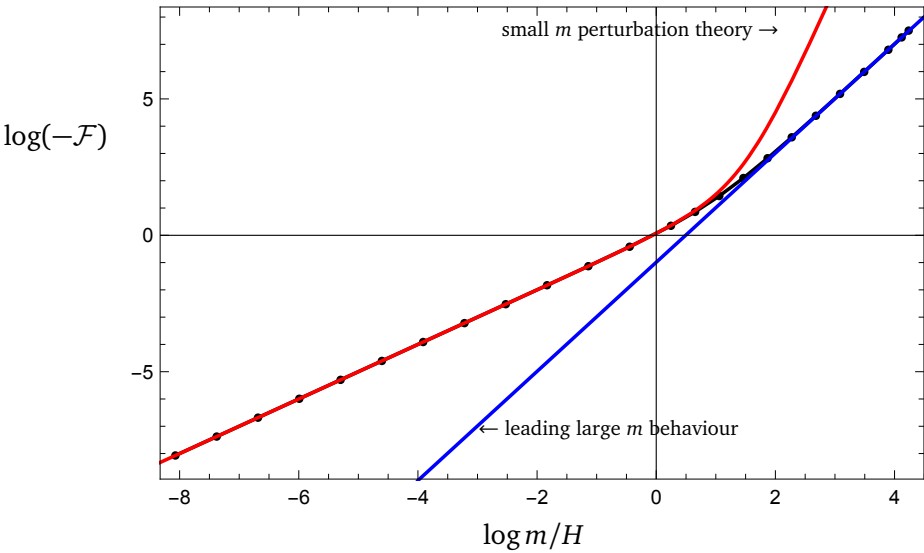

Figure 4: Asymptotic behaviour of the one-point functions as given by the function $\mathcal{F}$ appearing in (32) and (33) shown on a log-log plot. The red line gives the small $m$ behaviour as in (35) and the blue line gives the asymptotically large $m$ behaviour where the one-point functions are independent of $H$ as in (36). The black points are the numerical results at finite $m$.

## 4 Fluctuations and two-point functions

In order to compute correlation functions we first need to solve the equations of motion for linearised fluctuations on top of the domain wall background,

$$G_{ab} = G_{ab}^{\text{DW}}(z) + H_{ab}(z, x), \qquad \phi = \bar{\phi}(z) + H_\phi(z, x), \tag{37}$$

where $G_{ab}^{\text{DW}}(z), p(z)$ are given in (4). These fluctuations must obey suitable regularity conditions in the bulk and be consistent with the choice of vacuum. Once solved, we may read

---

[2]This procedure is not dissimilar to taking the planar limit of a black hole in global AdS by parametrically suppressing the curvature scale of the sphere. Here it is the de Sitter expansion which is suppressed.

off the corresponding sources $h_{(0)\mu\nu}, h_{\phi(1)}$ by moving to Fefferman-Graham gauge, and in particular their effect on the near-boundary data required to evaluate normalisable modes and two-point functions.

This calculation is facilitated by an appropriate spin-decomposition of the fluctuations,

$$
\begin{align}
H_{zz} &= X\,, && (38) \\
H_{z\mu} &= P(z)\big(\partial_\mu V + V_\mu\big)\,, && (39) \\
H_{\mu\nu} &= P(z)\Big(-2\psi g_{\mu\nu}^{dS} + 2\nabla_{(\mu}^{dS}\partial_{\nu)}\chi + 2\nabla_{(\mu}^{dS}\omega_{\nu)} + \gamma_{\mu\nu}\Big)\,, && (40) \\
H_\phi &= S\,, && (41)
\end{align}
$$

where $\gamma_{\mu\nu}$ is transverse traceless and $\omega_\mu, V_\mu$ are divergence free with respect to $g_{\mu\nu}^{dS}$.[3] The functions appearing here, $X, V, V_\mu, \psi, \chi, \omega_\mu, \gamma_{\mu\nu}, S$ each depend on all coordinates, $z, t, x^i$. There is gauge-dependence which can be reached by performing linearised diffeomorphisms by a vector $\xi_a$, whose overall effect is to adjust the metric and scalar perturbations by Lie derivatives of the fields,

$$
H_{ab} \to H_{ab} + 2\nabla_{(a}\xi_{b)}\,, \qquad H_\phi \to H_\phi + \xi^a \partial_a \bar{\phi}\,. \tag{42}
$$

A convenient way to proceed is to partially fix this gauge dependence by choosing the fluctuations to obey $H_{zz} = H_{z\mu} = 0$ which makes later comparison to Fefferman-Graham gauge straightforward. This corresponds to $X = V = V_\mu = 0$. There is then a residual gauge redundancy which preserves this choice, given by $\xi_a$ obeying,

$$
\begin{align}
\partial_z \xi_z &= 0\,, && (43) \\
\partial_\mu\left(\frac{\xi_z}{P}\right) + \partial_z\left(\frac{\xi_\mu}{P}\right) &= 0\,, && (44)
\end{align}
$$

for which we can write the most general solution,

$$
\xi_z(z,x) = f_z(x)\,, \qquad \xi_\mu(z,x) = P(z)\left(f_\mu(x) - \partial_\mu f_z(x)\int^z \frac{dq}{P(q)}\right)\,. \tag{45}
$$

Among the remaining variables, performing this residual transformation corresponds to adjusting the set of metric fluctuations. We further decompose $\xi_\mu = Z_\mu + \partial_\mu Z$ such $Z_\mu$ is divergence-free and further discuss the tensors, vectors and scalars separately in their respective sections 4.1, 4.2, 4.3 below.

In all cases it will become convenient to further decompose the fluctuations $\Phi \in \{X, V, V_\mu, \psi, \chi, \omega_\mu, \gamma_{\mu\nu}, S\}$ via separation of variables according to the de Sitter symmetries present in the background domain wall. We choose to do this utilising commuting labels corresponding to the two translation generators, $\frac{\partial}{\partial y^i}$ ($i = 1, 2$), and the Casimir operator $\Box_{dS_3}$, such that

$$
\partial_j \Phi = ik_j \Phi\,, \quad \Box_{dS_3}\Phi = \lambda\Phi\,. \tag{46}
$$

Under separation of variables keeping the regular solution this leads to fluctuations of the form[4]

$$
\Phi = \Phi_{k,\lambda}(z)\,\eta J_\nu(k\eta)\,e^{ik_i y^i}\,, \tag{47}
$$

where the Bessel index $\nu$ is determined by $\lambda$ through $\lambda = H^2\big(1 - \nu^2\big)$. This separation results in the bulk fluctuation equations becoming ODEs in domain wall radial variable $z$ only, with $\lambda, k_i$ appearing as parameters.

---

[3]This ansatz is the decomposition adopted in [45] generalised to a $dS_3$ boundary metric.

[4]The other eigenfunction of $\Box_{dS_3}$, $\eta Y_\nu(k\eta)$ diverges.

## 4.1 Tensors: $\gamma_{\mu\nu}$

The transverse traceless tensor $\gamma_{\mu\nu}$ is unaffected by the residual gauge transformations described above and is automatically gauge invariant. The most general form of $\gamma_{\mu\nu}$ is given as follows,

$$\gamma_{00} = 0, \tag{48}$$

$$\gamma_{0i} = -h_i J_\nu(k\eta) e^{ik\cdot y} \gamma(z), \tag{49}$$

$$\gamma_{ij} = \frac{1}{k^2\eta^2}(\eta\partial_\eta - 2)(\eta J_\nu(k\eta))\partial_{(i} e^{ik\cdot y} h_{j)} \gamma(z), \tag{50}$$

where $h_i$ is a constant polarisation 2-vector satisfying $h_i k^i = 0$. The field $\gamma(z)$ obeys the following radial equation of motion,

$$\gamma'' + \frac{3P'}{2P}\gamma' - \frac{\lambda}{P}\gamma = 0, \tag{51}$$

which we solve order-by-order in $m$, expanding $\lambda = \sum_{k=0} \lambda_k \left(\frac{m}{H}\right)^k$, $\gamma(z) = \sum_{k=0} \gamma_k(z)\left(\frac{m}{H}\right)^k$ alongside the expansion of $P$, subject to regularity in the bulk. From this solution we may read off the asymptotic data near the AdS boundary and extract the fluctuation $\gamma$'s contribution to $g_{(3)\mu\nu}$ and $g_{(0)\mu\nu}$ (see Appendix B for details). Using the expression for the one-point function in the presence of sources, (125), we may perform a variation of $\langle T_{\mu\nu}\rangle$ with respect to this transverse traceless fluctuation and recover the appropriate two-point function,

$$\langle T_{\mu\nu}(\nu_1, k_1) T_{\rho\sigma}(\nu_2, k_2)\rangle = \Pi_{\mu\nu\rho\sigma}(\nu_1, k_1; \nu_2, k_2) \mathcal{A}(\nu_1, k_1), \tag{52}$$

where $\Pi_{\mu\nu\rho\sigma}$ is a projection tensor which encodes the variation for transverse traceless perturbations,

$$\Pi_{\mu\nu}{}^{\rho\sigma}(\nu_1, k_1; \nu_2, k_2) = \frac{\delta g_{(0)\mu\nu}^{\mathrm{TT}}(\nu_1, k_1)}{\delta g_{(0)\rho\sigma}^{\mathrm{TT}}(\nu_2, k_2)}. \tag{53}$$

The projector (53) encodes conservation of spatial momenta and is also diagonal in Bessel index, proportional to $\delta_{\nu_1\nu_2}$. To illustrate the structure of the remaining amplitude $\mathcal{A}$, we present it to order $m^4$,

$$\begin{aligned}
\mathcal{A}(\nu, k) = \frac{H^3}{2\kappa^2} \Bigg[ & \nu(\nu^2 - 1) + \frac{3\nu^2 + 8\nu - 19}{24(\nu - 2)}\frac{m^2}{H^2} \\
& + \left(\frac{35}{864} - \frac{23}{1536(\nu-2)} + \frac{3}{256(\nu-2)^2} + \frac{1}{18(\nu-3)} + \frac{25}{512(\nu-4)}\right)\frac{m^4}{H^4} \\
& + O\left(\frac{m}{H}\right)^6 \Bigg].
\end{aligned} \tag{54}$$

The most salient observation here is that (54) contains poles appearing at specific integer values of the Bessel index, $\nu = 2, 3, \ldots$. Not only simple poles, but higher order poles too. The higher order poles appearing in this $m$ expansion are symptomatic of the simple pole locations receiving perturbative corrections order-by-order in $m$. These corrected simple pole locations correspond precisely to normalisable modes in the bulk which can be computed directly by setting fluctuations of the source to zero and imposing bulk regularity. This fixes a discrete spectrum of modes, reminiscent of a quasinormal mode calculation in a black hole background. See Appendix E for further details. This calculation reveals modes located at $\nu = \nu_n^t = n + O(m)$

for $n = 2, 3, \ldots$. The first few modes are given as follows,

$$v_2^t = 2 + \frac{1}{32}\frac{m^2}{H^2} - \frac{103}{36864}\frac{m^4}{H^4} + \frac{50929}{212336640}\frac{m^6}{H^6} + O(m)^8, \tag{55}$$

$$v_3^t = 3 + \frac{1}{864}\frac{m^4}{H^4} - \frac{49}{311040}\frac{m^6}{H^6} + O(m)^8, \tag{56}$$

$$v_4^t = 4 + \frac{5}{12288}\frac{m^4}{H^4} - \frac{119}{5308416}\frac{m^6}{H^6} + O(m)^8, \tag{57}$$

$$v_5^t = 5 + \frac{13}{518400}\frac{m^6}{H^6} + O(m)^8, \tag{58}$$

$$v_6^t = 6 + \frac{35}{4718592}\frac{m^6}{H^6} + O(m)^8, \tag{59}$$

and to the order we are working we also require $v_j^t = j + O(m)^8$ for $j = 7, 8$. With these modes known, we find the following spectral representation of the two point function, after a suitable resummation to incorporate the corrected mode locations,

$$\mathcal{A}(v, k) = \frac{3H^3}{2\kappa^2}\left(\frac{v}{3}(v^2 - 1) + \frac{v}{24}\frac{m^2}{H^2} + \sum_{j=2}^{\infty}\frac{r_j^t}{v - v_j^t}\right) - \frac{7}{12}m\langle O\rangle_0, \tag{60}$$

where the expectation value $\langle O\rangle_0$ given by in section 3 (which takes the perturbative expansion (35)) captures all corrections to $\mathcal{A}$ that are independent of $v$. Thus it takes the form of a contact term multiplied by a projection operator, and as such it is reminiscent of a Dilaton pole, see section 5.1 of [46]. The residues $r_j^t$ are determined order-by-order in this procedure and we present the first few in Appendix C.

## 4.2   Vectors: $\omega_\mu$

Among the vectors the residual diffeomorphisms act to transform the perturbation variables as follows:

$$\omega_\mu \;\rightarrow\; \omega_\mu + P^{-1}Z_\mu, \tag{61}$$

where $P^{-1}Z_\mu = f_\mu(x)$ by (45), hence an appropriate gauge invariant variable is simply

$$\hat{v}_\mu = \omega_\mu'. \tag{62}$$

For functions with arbitrary dependence on $(t, x, y)$ the momentum constraints reveal that

$$\hat{v}_\mu = 0, \tag{63}$$

trivialising the vector perturbations in $d = 3$.

## 4.3   Scalars: $\psi, \chi, S$

Among the scalars the residual diffeomorphisms act to transform the perturbation variables as follows:

$$\psi \;\rightarrow\; \psi + \frac{P'}{2P}\xi_z, \tag{64}$$

$$\chi \;\rightarrow\; \chi + \frac{1}{P}Z, \tag{65}$$

$$S \;\rightarrow\; S + \bar{\phi}'\xi_z. \tag{66}$$

From this we can identify gauge invariant combinations,

$$\zeta = -\psi + \frac{P'}{2P}\frac{S}{\bar{\phi}'}, \tag{67}$$

$$\hat{\phi} = -\left(\frac{S}{\bar{\phi}'}\right)', \tag{68}$$

$$\hat{v} = \chi' + \frac{S}{P\bar{\phi}'}, \tag{69}$$

where in constructing $\hat{\phi}$ we utilised (43) and constructing $\hat{v}$ we utilised (44). The Hamiltonian and momentum constraint equations give,

$$\hat{\phi} = \frac{2PH^2}{P'}\hat{v} - \frac{2P}{P'}\zeta', \qquad \hat{v} = -\frac{2(3H^2+\lambda)P'}{Q_\lambda}\zeta + \frac{Q_{-3H^2}}{H^2 Q_\lambda}\zeta', \tag{70}$$

where for convenience we have defined the background quantity $Q_\alpha(z) \equiv 12H^4 P - 2H^2 P^2(6+\bar{\phi}^2) - \alpha(P')^2$. Thus, once $\zeta$ is determined all other gauge invariant variables are determined by the above relations. All equations of motion are satisfied once $\zeta$ obeys the following ODE in the bulk radial direction, $z$,

$$\zeta'' + \left(\frac{Q'_{-3H^2}}{Q_{-3H^2}} - \frac{Q'_\lambda}{Q_\lambda} - \frac{12H^4 P'}{Q_{-3H^2}} + \frac{3P'}{2P}\right)\zeta' \tag{71}$$

$$-H^2\left(\frac{Q_\lambda(Q_\lambda + 3(3H^2+\lambda)(P')^2) + 2(3H^2+\lambda)P(-P'Q'_\lambda + Q_\lambda(2H^2 + P''))}{PQ_{-3H^2}Q_\lambda}\right)\zeta = 0.$$

Equivalently, we may use the following second order differential equation for $\hat{\phi}(z)$,

$$\hat{\phi}'' + \left(-\frac{4\bar{\phi}}{\bar{\phi}'} + \frac{2H^2}{P'} - \frac{2P}{P'} - \frac{\bar{\phi}^2 P}{3P'} - \frac{P\bar{\phi}'^2}{6P'}\right)\hat{\phi}' \tag{72}$$

$$+ \left(-10 - \bar{\phi}^2 - \frac{8\bar{\phi}^2}{\bar{\phi}'^2} + \frac{40H^2\bar{\phi}}{\bar{\phi}'P'} - \frac{40\bar{\phi}P}{\bar{\phi}'P'} - \frac{20\bar{\phi}^3 P}{3\bar{\phi}'P'} - \frac{10\bar{\phi}P\bar{\phi}'}{3P'} - \frac{\lambda}{P}\right)\hat{\phi} = 0,$$

which we solve by expanding order-by-order in $m/H$: $\lambda = \sum_{k=0}\lambda_k\left(\frac{m}{H}\right)^k$, $\hat{\phi}(z) = \sum_{k=0}\hat{\phi}_k(z)\left(\frac{m}{H}\right)^k$ alongside an expansion of the background fields $\bar{\phi}, P$. The equation of motion (72) gives rise to two integration constants, one corresponding to the choice of source and one determined through regularity. The physical fields are then uniquely determined through (70) and (67), (68), (69) up to gauge transformations completing the calculation of the scalar two point functions. Consider the variation of the action with respect to the scalar sources,

$$\delta S_{\text{scalars}} = \int d^3x \sqrt{g_{(0)}}\left(\frac{1}{2}\langle T_{\mu\nu}\rangle \delta g^{\mu\nu}_{(0)} + \langle O\rangle \delta\phi_{(1)}\right) \tag{73}$$

$$= \int d^3x \sqrt{g_{(0)}}\langle O\rangle\left(S_{(1)} + m\psi_{(0)}\right) = -m\int d^3x \sqrt{g_{(0)}}\langle O\rangle\zeta_{(0)}, \tag{74}$$

with $\langle O\rangle = \langle O\rangle_0 + S_{(2)}$ and where here it is understood that $\delta g^{\mu\nu}_{(0)}$ corresponds only to the scalar pieces. That is to say, the piece of the source fluctuation which implements a Weyl transform does not appear as guaranteed by the trace Ward identity, leaving only the linear combination $\zeta_{(0)}$. Hence

$$\langle O(x_1)O(x_2)\rangle = \frac{-1}{m}\frac{1}{\sqrt{g_{(0)}}}\frac{\delta S_{(2)}(x_1)}{\delta\zeta_{(0)}(x_2)}\bigg|_{\zeta_{(0)}=0}, \tag{75}$$

or, for our decomposition into Bessel functions,

$$\left\langle O_{\nu_1}(k_1) O_{\nu_2}(k_2) \right\rangle = \frac{S_{(2)}}{S_{(1)} + m\psi_{(0)}} \delta_{\nu_1\nu_2} \delta^{(2)}(k_1 + k_2). \tag{76}$$

We find for the first few orders

$$
\begin{aligned}
\langle\!\langle O_\nu(k)O_\nu(-k)\rangle\!\rangle &= H\Bigg[\nu + \left(\frac{1}{6(\nu-1)} + \frac{5}{54(\nu-2)}\right)\frac{m^2}{H^2} \\
&+ \left(-\frac{1}{24(\nu-1)} + \frac{1}{72(\nu-1)^2} + \frac{2017}{27648(\nu-2)} + \frac{19}{1152(\nu-2)^2}\right. \\
&\left. -\frac{5}{192(\nu-2)^3} - \frac{1}{36(\nu-3)} - \frac{71}{13824(\nu-4)} + \frac{5}{3072(\nu-6)}\right)\frac{m^4}{H^4} \\
&+ O\left(\frac{m}{H}\right)^6\Bigg],
\end{aligned} \tag{77}
$$

where the double angle brackets indicate we have dropped the Kronecker and Dirac delta factors.

As in the tensor case, the $m$-expanded scalar two point function contains poles (both single and higher order) at integer values of the Bessel index $\nu$. These correspond to the leading behaviour of normalisable modes where sources are turned off and regularity imposed in the bulk. Performing an explicit calculation to compute the mode spectrum, we find they are located at $\nu = \nu_n^s = n + O(m)$ where $n \in \mathbb{Z}^+$. Our computation of these modes extends the analogous calculation performed in [27] to finite $k$.[5] The first few modes are given by

$$
\nu_1^s = 1 + \frac{1}{12}\frac{m^2}{H^2} - \frac{1}{54}\frac{m^4}{H^4} + \frac{1591}{622080}\frac{m^6}{H^6} + O(m)^8, \tag{78}
$$

$$
\nu_{2,\pm}^s = 2 \mp \frac{i\sqrt{2}}{4}\frac{m}{H} + \frac{11}{192}\frac{m^2}{H^2} \mp \frac{37i\sqrt{2}}{12288}\frac{m^3}{H^3} + \frac{1855}{221184}\frac{m^4}{H^4} \pm \frac{2076503 i\sqrt{2}}{1132462080}\frac{m^5}{H^5} + O(m)^6, \tag{79}
$$

$$
\nu_3^s = 3 - \frac{1}{216}\frac{m^4}{H^4} + \frac{337}{777600}\frac{m^6}{H^6} + O(m)^8, \tag{80}
$$

$$
\nu_4^s = 4 - \frac{5}{4096}\frac{m^4}{H^4} - \frac{589}{15925248}\frac{m^6}{H^6} + O(m)^8, \tag{81}
$$

$$
\nu_5^s = 5 - \frac{37}{1036800}\frac{m^6}{H^6} + O(m)^8, \tag{82}
$$

$$
\nu_6^s = 6 - \frac{35}{2359296}\frac{m^6}{H^6} + O(m)^8, \tag{83}
$$

and to the order we are working we also require $\nu_j^s = j + O(m)^8$ for $j = 7, 8, 10, 12, 14$. Again, as with the tensor calculation, knowing the location of these poles order-by-order in $m$ allows a suitable resummation of the scalar two point function (77) such that it is entirely expressible as a sum over simple poles located at the mode indices $\nu_j^s$ with residues $r_j^s$,

$$
\langle\!\langle O_\nu(k)O_\nu(-k)\rangle\!\rangle = H\left(\nu + \frac{r_1^s}{\nu - \nu_1^s} + \sum_\pm \frac{r_{2,\pm}^s}{\nu - \nu_{2,\pm}^s} + \sum_{j=3}^\infty \frac{r_j^s}{\nu - \nu_j^s}\right). \tag{84}
$$

Note the lack of higher order poles. This procedure uniquely determines the residues $r_j^s$ order-by-order in $m$, the first few of which can be found in Appendix C.

---

[5]The corresponding frequencies given in [27] are obtained through $\omega_{\text{there}} = -i(\nu+1)H$, which can be seen at small $\eta$ or $k$ since $\eta J_\nu(k\eta) \sim \eta^{\nu+1} \sim e^{-(\nu+1)Ht}$.

Finally we note that there are two additional scalar two-point functions, $\langle T^{\mu}_{\mu} O \rangle$ and $\langle T^{\mu}_{\mu} T^{\nu}_{\nu} \rangle$ which follow from Ward identities given in (126). In particular taking further variations of the trace Ward identity gives $\langle T^{\mu}_{\mu} O \rangle = -\langle O \rangle_0$ and $\langle T^{\mu}_{\mu} T^{\nu}_{\nu} \rangle = 0$.

## 5 A Bessel function decomposition of CFT two-point functions

In this paper we have obtained expressions for two-point functions perturbatively in $m$ – see (84) for scalars and (60) for tensors. As previously discussed, these are naturally expressed using spatial wavevector $k^i$ and a Bessel index $\nu$ leading to the separation of variables $\sim \eta J_\nu(k\eta) e^{ik \cdot y}$. The leading $m = 0$ terms in (84) and (60) correspond to the CFT result, and as such represent a novel decomposition of standard CFT two-point functions. The goal of this section is to demonstrate this decomposition explicitly; starting with two point functions expressed in the Bessel function basis, we perform a change of basis and recover standard momentum-space expressions for CFT two-point functions, namely, for a scalar operator of dimension $\Delta$ in $d$-dimensional Minkowski spacetime [47,48],[6]

$$\langle O(\omega, k) O(\omega', k') \rangle = |k^2 - \omega^2|^{\Delta - \frac{d}{2}} \delta(\omega + \omega') \delta^{(d-1)}(k + k'). \tag{85}$$

First, we generalise our result for the scalar two-point function (84) to general $\Delta, d$, working at $m = 0$. Consider a scalar operator of dimension $\Delta$, $O_\phi$, with a source function $\phi_{(d-\Delta)}$ in $dS_d$. We set the source $\phi_{(d-\Delta)}$ to be an eigenfunction of the type discussed above, and via a probe holographic calculation imposing regularity, read off the resulting vev in the presence of the source, $\langle O_\phi \rangle_{\phi_{(d-\Delta)}}$. After Weyl transforming to Minkowski space, $-d\tau^2 + dy^2_{d-1}$, we find,[7]

$$\phi_{(d-\Delta)}(\nu) = \tau^{\frac{\delta-1}{2}} J_\nu(k\tau) e^{ik \cdot y},$$

$$\langle O_\phi(\nu) \rangle_{\phi_{(d-\Delta)}} = -\frac{2^{-\delta} e^{-i\pi\delta} \Gamma\left(1 - \frac{\delta}{2}\right) \Gamma\left(\frac{1}{2} - n + \frac{\delta}{2}\right)}{\Gamma\left(1 + \frac{\delta}{2}\right) \Gamma\left(\frac{1}{2} - n - \frac{\delta}{2}\right)} \tau^{-\frac{\delta+1}{2}} J_\nu(k\tau) e^{ik \cdot y}, \tag{86}$$

where $\delta = 2\Delta - d$. When $\Delta = 2, d = 3, (\delta = 1)$ this reduces to the result considered earlier in this paper, i.e. the $m = 0$ limit of (84).

Next, we wish to change the basis so that all $\tau$ dependence appears as a plane wave, $e^{-i\omega\tau}$, by summing (86) over the index $\nu$ with appropriately chosen coefficients. This is a simple exercise when $\delta = 1$, due to the following generating function for $J_\nu(x)$,

$$e^{\frac{x}{2}\left(t - \frac{1}{t}\right)} = \sum_{n=-\infty}^{\infty} t^n J_n(x), \tag{87}$$

which informs the correct choice of coefficients when summing the source term in (86) to obtain,

$$\sum_{\nu=-\infty}^{\infty} t^\nu \phi_{(d-\Delta)}(\nu) = e^{ik \cdot y - i\omega\tau}, \qquad \text{where} \quad t - \frac{1}{t} = -2i\frac{\omega}{k}. \qquad (\delta = 1) \tag{88}$$

Applying the same sum to the vev, we obtain,

$$\sum_{\nu=-\infty}^{\infty} t^\nu \langle O_\phi(\nu) \rangle_{\phi_{(d-\Delta)}} = \sum_{\nu=-\infty}^{\infty} -\frac{\nu}{\tau} t^\nu J_\nu(k\tau) e^{ik \cdot y} \qquad (\delta = 1)$$

$$= -\frac{t}{\tau} \partial_t \left( e^{\frac{k\tau}{2}\left(t - \frac{1}{t}\right)} \right) e^{ik \cdot y} = \pm \sqrt{k^2 - \omega^2} e^{ik \cdot y - i\omega\tau}, \tag{89}$$

---

[6]Reached from de Sitter by a Weyl transformation.

[7]Here and in appendix D we have set $2\kappa^2 = 1$.

thus completing the derivation of the standard CFT two-point function in momentum space (85) at $\delta = 1$, up to normalisation factors. In appendix D we extend this result to all odd $\delta$, which requires several additional steps due to the powers of $\tau$ appearing in the source function.

## 5.1 Mass corrections

In the presence of the mass deformation, $m$, we may go beyond the CFT result and compute corrections to (85). As shown in both the tensor (60) and scalar (84) two-point function calculations, corrections in $m$ lead to the addition of simple poles at non-integer values of the Bessel index (after suitable resummation of the perturbative expansion). Let us consider one such correction of this type, which we denote by the sum $S$, and compute its contribution to the momentum space result in flat space. The general correction to the flat-space vev of interest takes the form,

$$S = \sum_{n=-\infty}^{\infty} a_n t^n \tau^{-1} J_n(k\tau), \tag{90}$$

for a source with frequency and momentum $\omega, k^i$, where $a_n$ are independent of $\tau$. We now perform the Fourier transform of this expression from $\tau$ to $\omega'$ which gives us the contribution to the two-point function describing the response at $\omega'$ for a source at $\omega$. At $m = 0$ the two-point function was proportional to $\delta(\omega + \omega')$ but this is no longer the case at $m \neq 0$ since the deformation breaks time-translation invariance. Restricting for concreteness to $0 < \omega' < k$ the Fourier transform is given by,

$$\tilde{S} = -i \sum_{n=-\infty}^{\infty} a_n t^n \left( \frac{\sin(n\pi)}{2\pi n} - \frac{i^n \sin\left(n\left(\frac{\pi}{2} + \psi'\right)\right)}{n\pi} \right), \tag{91}$$

where $\psi'$ is an angle. This angle arises since we have the quantity $t' = \frac{-i\omega' + \sqrt{k^2 - (\omega')^2}}{k}$ which for the conditions $0 < \omega' < k$ is unit norm and thus may be represented by $-\frac{\pi}{2} < \psi' < 0$ where $t' = e^{i\psi'}$. For a simple pole at $n_0$, i.e. $a_n = \frac{1}{n - n_0}$, the sum (91) can be evaluated directly,

$$
\begin{aligned}
2\pi n_0 \tilde{S} &= \left( \frac{t}{t'} \right)^{n_0} \left[ B\left( \frac{t}{t'}; 1 - n_0, 0 \right) - B\left( \frac{t'}{t}; 1 + n_0, 0 \right) \right] \\
&\quad + (-t t')^{n_0} \left[ B\left( \frac{-1}{t t'}; 1 + n_0, 0 \right) - B\left( -t t'; 1 - n_0, 0 \right) \right] + i\pi,
\end{aligned} \tag{92}
$$

where $B(z; x, y)$ is the incomplete Euler beta function and $t = \frac{-i\omega + \sqrt{k^2 - \omega^2}}{k}$.

## 6 Free fermion

A mass deformation in $dS_3$ corresponds to a $\Delta = 2$ operator, which we denoted above as $O$. In three dimensions this is the dimension of a mass term for a fermion. In this section we consider a massive free fermion theory on $dS_3$ to assess which features of the above analysis are sensitive to the coupling strength and which are intrinsic to massive theories placed on $dS_3$.

We begin with the two-, three- and four-point functions for $O = \bar{\psi}\psi$ where $\psi$ is a Euclidean

free massless two-component fermion on $\mathbb{R}^3$. As it is a CFT, these take the general form,

$$\langle O(x_1)O(x_2)\rangle_0 = \frac{\alpha}{|x_{12}|^4}, \tag{93}$$

$$\langle O(x_1)O(x_2)O(x_3)\rangle_0 = \frac{\alpha C}{|x_{12}|^2 |x_{23}|^2 |x_{31}|^2}, \tag{94}$$

$$\langle O(x_1)O(x_2)O(x_3)O(x_4)\rangle_0 = \frac{1}{x_{12}^4 x_{34}^4} f(u,v), \tag{95}$$

where the subscript 0 indicates that these are computed for the massless theory. Appearing in these expressions are the conformally invariant cross ratios

$$u = \frac{x_{12}^2 x_{34}^2}{x_{13}^2 x_{24}^2}, \qquad v = \frac{x_{14}^2 x_{23}^2}{x_{13}^2 x_{24}^2}. \tag{96}$$

We evaluate the remaining data $\alpha, C, f$ by direct calculation in position space. The propagator is given by

$$\langle \psi(x_1)\bar{\psi}(x_2)\rangle = \frac{i\slashed{x}_{12}}{4\pi|x_{12}|^2}. \tag{97}$$

Each $O$ insertion introduces a single factor of the Dirac matrices $\gamma^\mu$. For the two point function using $\text{tr}(\gamma^\mu\gamma^\nu) = 2\delta^{\mu\nu}$ we recover the normalisation $\alpha = 1/(8\pi^2)$. For the three point function the result can only depend on two differences $x_{12}, x_{23}$ with the third determined $x_{31} = -x_{12} - x_{23}$. Since $\text{tr}(\gamma^\mu\gamma^\nu\gamma^\rho) = 2\epsilon^{\mu\nu\rho}$ we can see that the three point function must therefore vanish since there are not enough linearly independent variables to construct a nonzero contraction with $\epsilon^{\mu\nu\rho}$. Thus $C = 0$. For the four point function, a more detailed calculation is required. There are 4 contractions for each diagram that contributes to $\langle O(x_1)O(x_2)O(x_3)O(x_4)\rangle_0$ and 6 total diagrams. It is useful to note the identity $\text{tr}(\gamma^\mu\gamma^\nu\gamma^\rho\gamma^\sigma) = 2(\delta^{\mu\nu}\delta^{\rho\sigma} - \delta^{\mu\rho}\delta^{\nu\sigma} + \delta^{\mu\sigma}\delta^{\nu\rho})$ used in evaluating the final trace. Summing all diagrams gives the final result,

$$f(u,v) = \alpha^2 \frac{u^{1/2}}{2v^{3/2}}\left(1 - u - v - u^{3/2} - v^{3/2} + u^{5/2} + v^{5/2} - uv^{3/2} - vu^{3/2}\right). \tag{98}$$

As a check one may verify that this expression obeys the required crossing identities,

$$\left(\frac{v}{u}\right)^2 f(u,v) = f(v,u), \qquad f(u,v) = f\left(\frac{u}{v}, \frac{1}{v}\right). \tag{99}$$

The above expressions (93), (94), (95) can be used within conformal perturbation theory to evaluate two point functions for the massive theory. Consider a conformal field theory $S_{\text{CFT}}$ deformed by a scalar operator $O$ of dimension $\Delta$ with spacetime-dependent coupling

$$S = S_{\text{CFT}} + \int d^d x\, m(x)O(x). \tag{100}$$

Then a correlation function in the full theory can be constructed by computing CFT correlation functions with the following insertion,

$$e^{-\int d^d x\, m(x)O(x)} = \sum_{n=0}^{\infty} \frac{(-1)^n}{n!} \int d^d x_1 \ldots d^3 x_n \left(m(x_1)O(x_1)\ldots m(x_n)O(x_n)\right). \tag{101}$$

Let us first consider the leading correction to the one point function,

$$\langle O(x_1)\rangle = 0 - \int d^3 x_2 m(x_2) \langle O(x_1)O(x_2)\rangle_0 + O(m)^2. \tag{102}$$

First performing the two spatial integrals we obtain,

$$\langle O(x_1)\rangle = \frac{im}{8\pi H}\int d\tau_2 \frac{1}{\tau_2}\frac{1}{\tau_{12}^2} + O(m)^2\,, \tag{103}$$

and hence choosing a contour for the $\tau_2$ integral which picks up the pole at $\tau_2 = 0$ we find

$$\langle O(x_1)\rangle = \frac{m}{4H\tau_1^2} + O(m)^2\,. \tag{104}$$

The final step is to perform a Weyl transformation with $\Omega = (Hi\tau)^{-1}$ and analytic continuation $\tau = i\eta$ to go from a theory on $g = d\tau^2 + d\vec{x}^2$ to a theory on dS$_3$ with line element $\Omega^2 g = (H^2\eta^2)^{-1}(-d\eta^2 + d\vec{x}^2)$,

$$\langle O\rangle_{\Omega^2 g} = -H^2\frac{1}{4}\frac{m}{H} + O(m)^2\,, \tag{105}$$

which matches the holographic result at this order (32) up to a constant.

Next we consider two point functions evaluated perturbatively in $m$ up to order $m^2$. We have,

$$
\begin{aligned}
\langle O(x_1)O(x_2)\rangle &= \langle O(x_1)O(x_2)\rangle_0 \\
&\quad - \int d^d x_3 m(x_3)\langle O(x_1)O(x_2)O(x_3)\rangle_0 \\
&\quad + \frac{1}{2}\int d^d x_3 d^d x_4 m(x_3)m(x_4)\langle O(x_1)O(x_2)O(x_3)O(x_4)\rangle_0\,.
\end{aligned}
\tag{106}
$$

At order $m^2$ there are a set of integrals to perform, explicitly given here by inserting the above expression (95) with the result (98). To order $m$,

$$\langle O(x_1)O(x_2)\rangle = \frac{1}{8\pi^2|x_{12}|^4} + O(m)^2\,. \tag{107}$$

Which matches the holographic result for the scalar two-point function, including the vanishing of the order $m$ term. In future work it would be interesting to evaluate the contribution of the data appearing in the four-point function through $f(u,v)$ which here appears at order $m^2$.

Finally, for completeness we also present the free fermion result (107) on de Sitter spacetime, labelled by spatial momenta $k_i$. Let $x = (\tau,\vec{x})$. Fourier transforming in $\vec{x}_1,\vec{x}_2$ gives

$$\left\langle O_{\vec{k}_1}(\tau_1)O_{\vec{k}_2}(\tau_2)\right\rangle = \frac{1}{32\pi^3}\frac{k}{\tau_{12}}K_1(k\tau_{12})\delta^{(2)}(\vec{k}_1 + \vec{k}_2) + O(m)^2\,, \tag{108}$$

where $K_1$ is a modified Bessel function. Finally we perform the Weyl transformation,

$$\left\langle O^L_{\vec{k}_1}(\eta_1)O^L_{\vec{k}_2}(\eta_2)\right\rangle_{\Omega^2 g} = -\frac{1}{32\pi^3}\frac{H^4\eta_1^2\eta_2^2}{\eta_{12}^2}(-ik\eta_{12}K_1(-ik\eta_{12}))\delta^{(2)}(\vec{k}_1 + \vec{k}_2) + O(m)^2\,, \tag{109}$$

where $O^L(\eta) \equiv O(i\eta)$. Note that the term in parenthesis goes to 1 in the limit $k \to 0$.

## 7  Discussion

In this work we have studied strongly-coupled non-conformal QFTs in fixed de Sitter backgrounds via holography. Non-conformality was introduced via a mass deformation of a CFT.

We computed scalar, vector and tensor two-point functions directly at strong coupling for a particular holographic model in $d = 3$.

One of our motivations was gaining a new perspective on correlation functions of massive quantum fields in fixed de Sitter backgrounds. By working directly at strong coupling one may hope to evade infrared issues which arise in perturbative approaches [1,2]. Our work is based around a deformed holographic CFT and when $m \to 0$ we recover standard CFT results in a controlled fashion. It would be interesting to revisit weak coupling calculations starting from the perspective of a CFT.

Through holography, the $dS_3$-invariant vacuum state for a massive theory is realised as an asymptotically-AdS$_4$ geometry [27]. Here, we have elucidated the global causal structure of this geometry. The mass deformation leads to a defect-like singular source on the boundary of AdS that propagates into the bulk forming a spacelike singularity. The spacetime is of domain-wall form, foliated by dS-invariant slices. This dS-invariant foliation already existed for the CFT case at $m = 0$ where the bulk is exactly AdS, but we have found that it is robust to adding the mass deformation and persists at finite $m \neq 0$ too. The warp-factor changes and the sliding freedom of the $m = 0$ foliation is pinned by the defect when $m \neq 0$. As a consequence, the dS-invariant slices naturally organise the holographic computation of correlation functions, where fluctuations sit in representations of the dS isometry algebra.

An interesting outcome of the two-point function calculations was their simplicity when expressed in a spectral representation as a sum over simple poles. These poles correspond to normalisable modes of the bulk spacetime. To see this emerge when working perturbatively in small $m$ requires an additional resummation step which corresponds to shifting the pole locations. The scalar normalisable modes had been computed before at $k = 0$ [27] and we have extended them to $k \neq 0$ and also computed tensor modes. The vector modes were trivial. When $m = 0$ the 2-point functions take the expected CFT form and our ability to resum the small $m$ results in pole shifts provides evidence that the deformed theory is stable and smoothly limits to the CFT as $m \to 0$. This can be seen in the late time expansion where the Bessel function with a shifted index corresponds to a decaying fluctuation as a power-law in conformal time.

While it is tempting to think of these modes as akin to QNMs of black holes, we have shown that their $k$-dependence does not contain any non-trivial dispersive information; the $k$ dependence is entirely determined by the dS-isometries present in the state. Thus we do not see any further analogies to hydrodynamic-like excitations as was enticingly conjectured in [27]. In particular, consider adding a normalisable mode fluctuation to the $dS_3$-invariant vacuum state. It has a rate of decay governed by the Bessel index for the fluctuation. We have shown that the spectrum of Bessel indices, akin to a QNM spectrum for a black hole, is independent of the spatial wavenumber of the fluctuation, $\vec{k}$. Moreover, the bulk mode functions themselves depend only on $\vec{k}$ in a way that is determined by symmetries. This is in stark contrast to hydrodynamics where the decay towards thermal equilibrium, as governed by a QNM, depends on $\vec{k}$ in a manner consistent with the conservation of energy and momentum.

There is another way to understand this, rooted in the isometries of the system. In the fluid-gravity setup [49], dilatations and boosts are isometries of AdS$_4$ that are broken by the black brane geometry. However, they are still asymptotic symmetries, and acting with these on the black brane geometry generate energy and velocity parameters which become hydrodynamic fields when promoted to slowly varying functions of spacetime [50]. The bulk solution in our case preserves 6 out of the maximum of 10 AdS$_4$ isometries (i.e. all of the $dS_3$ isometries). However, one can check that the 4 generators broken by the solution (two boosts $J_{0i}$, time translation $P_0$ and the special conformal transformation $K_0$) are also explicitly broken by the scalar field boundary conditions; *i.e.* they are not asymptotic symmetries. Thus the broken generators in the $dS_3$-invariant vacuum state cannot be used to generate additional parameters

of the solution, ruling out hydrodynamic-like behaviour.

Our work provides several interesting directions going forward. Here for simplicity we have focussed our attention on $dS_3$, but of course extending our work to correlation functions on $dS_4$ and making appropriate connections to cosmological observations is a desirable next step. To understand the role of strong coupling, in section 6 we set in motion a calculation of a free-fermion in conformal perturbation theory. The composite operator $\bar{\psi}\psi$ plays the role of the $\Delta = 2$ operator $O_\phi$ in the holographic calculation and it would be valuable to continue this calculation to higher orders in $m$ to identify which features are robust under changes in coupling strength.

Finally, and tangentially, the existence $dS_d$-foliation of $AdS_{d+1}$ at $m = 0$ leads to a novel basis in which to expressing $CFT_d$ correlation functions in Minkowski space. Fluctuations are plane waves in spatial directions and their time dependence are given by Bessel functions. We have explicitly provided a map between this representation and the standard momentum space expression. In future work it would be interesting to explore whether this (apparently simpler) Bessel basis provides computational conveniences or conceptual advantages for computations of CFT correlators in general.

## Acknowledgements

It is a pleasure to thank Alex Buchel and Enrico Parisini for discussions. JMP is supported in part by the Academy of Finland grant no. 1322307 and was supported by a Royal Society Research Fellow Enhancement Award during this work. KS and BW are supported in part by the Science and Technology Facilities Council (Consolidated Grant ST/T000775/1). BW is supported by a Royal Society University Research Fellowship.

## A  Ingoing coordinates

In [27] the following ansatz was utilised,[8]

$$ds^2 = \frac{-H dv dx - H^2 g(x) dv^2 + e^{2Hv} H^2 f(x)^2 d\vec{y}^2}{x^2}, \qquad \phi = p(x), \qquad (110)$$

with bulk equations of motion consisting of a second order equation for $f$ and $p$ with $g$ determined algebraically. In [27] bulk regularity was required for $x \in (0, x_{AH}]$ where $x_{AH}$ labels the location of an apparent horizon in the bulk, whose location is determined by the condition,[9]

$$\left(f(2x + g) - xgf'\right)\big|_{x=x_{AH}} = 0. \qquad (111)$$

It is a simple matter to show that the solutions constructed in these coordinates can be mapped to domain wall form (4) under the following coordinate transformation

$$x = X(z), \qquad v = t + \frac{1}{2H} \log\left(\frac{g(X(z))}{f(X(z))}\right). \qquad (112)$$

The resulting $dt dz$ cross terms vanishing courtesy of the bulk equations of motion leaving a metric of the form (4) with warp factor

$$P(z) = -\frac{H^2 g(X(z))}{X(z)^2}, \qquad (113)$$

---

[8]Here $m = Hp_1$ where $p_1$ is the deformation parameter defined in [27].
[9]This expression corrects a typo in (2.22) of [27].

where the remaining function $X(z)$ simply determines the domain wall choice of radial coordinate through,

$$(X'(z))^2 = X(z)^2 g(X(z)). \tag{114}$$

Finally we can comment on the apparent horizons and the regularity criterion adopted by [27] to enforce bulk regularity. There, regularity was required up to $x_{AH}$ given by (111). In the AdS coordinate extension, this surface corresponds to a fixed value $Y = Y_{AH}$, though note that the generator of this apparent horizon is not a radial geodesic and carries angular momentum. In particular, to leading order its location is given by

$$Y_{AH} = 1 - \frac{1}{6^{1/3}} \left(\frac{m}{H}\right)^{\frac{2}{3}} + O(m)^{4/3}. \tag{115}$$

Note that while $Y_{AH} < Y_*^-$ which ensures that it plays its desired role of delineating the region containing the singularity, it does not appear to carry any particular physical significance (as is to be expected from an apparent horizon).

## B  Holographic renormalisation

In order to compute one and two-point functions from holography, a consistent treatment of UV divergences is required [51–53]. Our starting point, as is standard, is to change coordinates so that the metric takes Fefferman-Graham form,

$$ds^2 = \frac{d\rho^2}{4\rho^2} + \frac{1}{\rho} g_{\mu\nu}(\rho, x) dx^\mu dx^\nu, \tag{116}$$

where $x^\mu$ are coordinates on dS$_3$. This only needs to be done near the boundary, where the bulk fields $g_{\mu\nu}, \phi$ take the following small $\rho$ expansions,

$$\phi(\rho, x) = \phi_{(1)}(x)\rho^{\frac{1}{2}} + \phi_{(2)}(x)\rho + O(\rho)^{\frac{3}{2}}, \tag{117}$$

$$g_{\mu\nu}(\rho, x) = g_{(0)\mu\nu}(x) + g_{(2)\mu\nu}(x)\rho + g_{(3)\mu\nu}(x)\rho^{\frac{3}{2}} + O(\rho)^2, \tag{118}$$

where we have explicitly included up to the required order where VEVs appear, and the labels indicate the scaling dimension of each term. For instance, the operator dual to the bulk field $\phi$ is dimension 2 and so we keep $O(\rho)$, while the boundary stress tensor is dimension 3 and so we keep $O(\rho)^{\frac{3}{2}}$. Among the data appearing here, $\phi_{(0)}$ and $g_{(0)}$ are source data while the equations of motion determine $g_{(2)}$ as

$$g_{(2)\mu\nu} = -R_{(0)\mu\nu} + \frac{1}{4} g_{(0)\mu\nu} R_{(0)} - \frac{1}{8} \phi_{(1)}^2 g_{(0)\mu\nu} \tag{119}$$

and place constraints on the remaining data $g_{(3)}, \phi_{(2)}$,

$$g_{(0)}^{\mu\nu} g_{(3)\mu\nu} = -\frac{2}{3} \phi_{(1)} \phi_{(2)}, \tag{120}$$

$$\nabla_{(0)}^\mu g_{(3)\mu\nu} = -\frac{1}{3} \phi_{(1)} \partial_\nu \phi_{(2)}. \tag{121}$$

Next we introduce a regulator at $\rho = \epsilon$ and analyse the structure of divergences that appear in the regulated action as $\epsilon \to 0$. The small $\epsilon$ expansion can be systematically inverted in terms of local terms intrinsic to the boundary, and give a divergent action $-S_{ct}$ where

$$S_{ct} = \frac{1}{2\kappa^2} \int \sqrt{-\gamma} \left( -R(\gamma) - 4 - \frac{1}{2}\phi^2 \right). \tag{122}$$

Subtracting these divergences from the regulated bulk action gives

$$S_{\text{sub}} = S_{\text{reg}} + S_{\text{ct}} . \tag{123}$$

with the renormalised action $S_{\text{ren}} = \lim_{\epsilon \to 0} S_{\text{sub}}$. Variations of the renormalised action give general expressions for the one point functions in the presence of sources,

$$\langle O \rangle = \frac{1}{\sqrt{g_{(0)}}} \frac{\delta S_{\text{ren}}}{\delta \phi_{(1)}} = \frac{1}{2\kappa^2} \phi_{(2)} , \tag{124}$$

$$\langle T_{\mu\nu} \rangle = \frac{2}{\sqrt{g_{(0)}}} \frac{\delta S_{\text{ren}}}{\delta g_{(0)}^{\mu\nu}} = -\frac{1}{2\kappa^2} \left( 3 g_{(3)\mu\nu} + g_{(0)\mu\nu} \phi_{(1)} \phi_{(2)} \right) . \tag{125}$$

Combining this with constraints on the near boundary data arising from bulk equations of motion (120), (121) give rise to the following trace and diffeomorphism Ward identities,

$$\left\langle T^{\mu}_{\ \mu} \right\rangle = -\phi_{(1)} \langle O \rangle , \qquad \nabla^{\mu} \left\langle T_{\mu\nu} \right\rangle = -\langle O \rangle \, \partial_\nu \phi_{(1)} . \tag{126}$$

We are now in a position to compute correlation functions by successive derivatives with respect to sources. In this paper we consider up to two point functions and for this purpose we examine solutions to the bulk equations of motion consisting of the background domain wall solution with metric and scalar fluctuations around it. In Fefferman-Graham form,

$$g_{\mu\nu}(\rho, x) = \bar{g}_{\mu\nu}(\rho, x) + h_{\mu\nu}(\rho, x) , \tag{127}$$

$$\phi(\rho, x) = \bar{\phi}(\rho) + h_\phi(\rho, x) , \tag{128}$$

where the background domain wall solution is labelled with a bar, whose source functions are

$$\bar{g}_{(0)\mu\nu} = g^{dS}_{\mu\nu} , \qquad \bar{\phi}_{(1)} = m . \tag{129}$$

In accordance with the use of Fefferman-Graham coordinates the fluctuations have a near-boundary form,

$$h_{\mu\nu}(\rho, x) = h_{(0)\mu\nu}(x) + h_{(2)\mu\nu}(x)\rho + h_{(3)\mu\nu}(x)\rho^{\frac{3}{2}} + O(\rho)^2 \tag{130}$$

$$h_\phi(\rho, x) = h_{\phi(1)}(x)\rho^{\frac{1}{2}} + h_{\phi(2)}(x)\rho + O(\rho)^{\frac{3}{2}} . \tag{131}$$

Here $h_{(0)\mu\nu}(x), h_{\phi(1)}(x)$ are recognised as linearised source functions, while $h_{(2)\mu\nu}(x)$ is determined by the near boundary equations of motion as in (119). These sources appear as Dirichlet boundary conditions to the bulk solution, and through the requirement of bulk regularity determine the remaining data as linear functionals of them, i.e. $h_{(3)\mu\nu}\left[h_{(0)\mu\nu}(x), h_{\phi(1)}(x)\right]$ and $h_{(2)\phi}\left[h_{(0)\mu\nu}(x), h_{\phi(1)}(x)\right]$. We thus arrive at the following formal expressions for the desired one-point functions,

$$\langle O(x) \rangle_0 = \frac{1}{\sqrt{\bar{g}_{(0)}}} \frac{\delta S_{\text{ren}}}{\delta h_{\phi(1)}(x)} \bigg|_{h_{\phi(1)} = h_{(0)\mu\nu} = 0} = \frac{1}{2\kappa^2} \bar{\phi}_{(2)} , \tag{132}$$

$$\langle T_{\mu\nu}(x) \rangle_0 = \frac{2}{\sqrt{\bar{g}_{(0)}}} \frac{\delta S_{\text{ren}}}{\delta h^{\mu\nu}_{(0)}(x)} \bigg|_{h_{\phi(1)} = h_{(0)\mu\nu} = 0} = -\frac{1}{2\kappa^2} \left( 3\bar{g}_{(3)\mu\nu} + \bar{g}_{(0)\mu\nu} \bar{\phi}_{(1)} \bar{\phi}_{(2)} \right) . \tag{133}$$

## C  Spectral decomposition residues

The residues appearing in the tensor two point function of (60) are as follows:

$$r_2^t = -\frac{m^2}{8H^2}\left(1 - \frac{23}{576}\frac{m^2}{H^2} - \frac{14477}{6635520}\frac{m^4}{H^4} + \frac{66506857}{133772083200}\frac{m^6}{H^6} + \dots\right), \quad (134)$$

$$r_3^t = -\frac{1}{54}\frac{m^4}{H^4}\left(1 - \frac{1}{9}\frac{m^2}{H^2} + \frac{607}{51840}\frac{m^4}{H^4} + \dots\right), \quad (135)$$

$$r_4^t = -\frac{25}{1536}\frac{m^4}{H^4}\left(1 - \frac{193}{4320}\frac{m^2}{H^2} - \frac{10541}{6635520}\frac{m^4}{H^4} + \dots\right), \quad (136)$$

$$r_5^t = -\frac{13}{6480}\frac{m^6}{H^6}\left(1 - \frac{53}{312}\frac{m^2}{H^2} + \dots\right), \quad (137)$$

$$r_6^t = -\frac{1225}{1179648}\frac{m^6}{H^6}\left(1 - \frac{88633}{907200}\frac{m^2}{H^2} + \dots\right), \quad (138)$$

$$r_7^t = -\frac{257}{1814400}\frac{m^8}{H^8}\left(1 + \dots\right), \quad (139)$$

$$r_8^t = -\frac{1225}{25165824}\frac{m^8}{H^8}\left(1 + \dots\right). \quad (140)$$

The residues appearing in the scalar two point function of (84) are as follows:

$$r_1^s = \frac{m^2}{6H^2}\left(1 - \frac{1}{4}\frac{m^2}{H^2} + \frac{109}{4536}\frac{m^4}{H^4} + \frac{109672267}{10059033600}\frac{m^6}{H^6} + \dots\right), \quad (141)$$

$$r_{2,\pm}^s = \frac{5}{48}\frac{m^2}{H^2}\left(1 \pm i\frac{7\sqrt{2}}{160}\frac{m}{H} + \frac{2017}{5760}\frac{m^2}{H^2} \pm i\frac{982129\sqrt{2}}{22118400}\frac{m^3}{H^3} - \frac{164027}{8847360}\frac{m^4}{H^4}\right.$$
$$\left.\pm i\frac{12471749309\sqrt{2}}{7927234560000}\frac{m^5}{H^5} - \frac{3680038951367}{535088332800000}\frac{m^6}{H^6} + \dots\right), \quad (142)$$

$$r_3^s = -\frac{1}{36}\frac{m^4}{H^4}\left(1 - \frac{7}{225}\frac{m^2}{H^2} + \frac{1543}{90720}\frac{m^4}{H^4} + \dots\right), \quad (143)$$

$$r_4^s = -\frac{71}{13824}\frac{m^4}{H^4}\left(1 + \frac{2317}{20448}\frac{m^2}{H^2} - \frac{1619171}{94224384}\frac{m^4}{H^4} + \dots\right), \quad (144)$$

$$r_5^s = -\frac{37}{103680}\frac{m^6}{H^6}\left(1 - \frac{103}{14504}\frac{m^2}{H^2} + \dots\right). \quad (145)$$

$$r_6^s = \frac{5}{3072}\frac{m^4}{H^4}\left(1 - \frac{1111}{14400}\frac{m^2}{H^2} + \frac{9812941}{995328000}\frac{m^4}{H^4} + \dots\right), \quad (146)$$

$$r_7^s = -\frac{4723}{362880000}\frac{m^8}{H^8}\left(1 + \dots\right), \quad (147)$$

$$r_8^s = \frac{101}{2419200}\frac{m^6}{H^6}\left(1 - \frac{14102555}{104251392}\frac{m^2}{H^2} + \dots\right), \quad (148)$$

$$r_{10}^s = \frac{25}{1572864}\frac{m^6}{H^6}\left(1 - \frac{7788127}{108864000}\frac{m^2}{H^2} + \dots\right), \quad (149)$$

$$r_{12}^s = \frac{55200281}{64377815040000}\frac{m^8}{H^8}\left(1 + \dots\right), \quad (150)$$

$$r_{14}^s = \frac{65}{301989888}\frac{m^8}{H^8}\left(1 + \dots\right). \quad (151)$$

# D  CFT two point functions for odd $\delta$

In this appendix we generalise a result obtained in the main text in section 5 from $\delta = 1$ to general odd $\delta = 1 + 2M$. This demonstrates recovery of standard CFT two point function in momentum space from an appropriate sum of Bessel functions.

We have the following pair of source, $s$, and vev, $v_M$, in the presence of source,

$$s = \tau^M \sum_{n=-\infty}^{\infty} c_n J_n(k\tau) e^{ik\cdot y}, \tag{152}$$

$$v_M = 2^{-1-2M} \tau^{-1-M} \frac{\Gamma(1/2-M)}{\Gamma(3/2+M)} \sum_{n=-\infty}^{\infty} c_n \frac{\Gamma(1+M-n)}{\Gamma(-M-n)} J_n(k\tau) e^{ik\cdot y}, \tag{153}$$

where the summation coefficients $c_n$ are to be determined in order to get a plane wave source function. The obstacle here compared with the $\delta = 1$ ($M = 0$) case is the appearance of $\tau^M$ in the source which prevents a direct application of the generating function (87). However, powers of $\tau$ can introduced into (87) to create a new generating function by applying the following identity,

$$J_n(k\tau) = \frac{k\tau}{2n} (J_{n-1}(k\tau) + J_{n+1}(k\tau)) \tag{154}$$

recursively $M$ times to generate $\tau^M$, giving,

$$e^{\frac{t^2-1}{2t}k\tau} = \sum_{n=-\infty}^{\infty} c_n \tau^M J_n(k\tau), \tag{155}$$

$$\text{with} \quad c_n(k) = \left(\frac{k}{2}\right)^M \sum_{q=0}^{M} \binom{M}{q} \frac{\Gamma(n-q)n}{\Gamma(n+M-q+1)} t^{n+M-2q}. \tag{156}$$

Thus, (156) is precisely the choice of $c_n$ we make in our summation to ensure the summed source in (153) becomes a plane wave, $s = e^{-i\omega\tau+ik\cdot y}$ where $\omega$ is given by $t, k$ as in (88). To find the two point function we now need the vev in the presence of this source under this sum. With (156) the vev becomes,

$$v_M = \left(\frac{k}{2}\right)^M 2^{-1-2M} \tau^{-1-M} \tag{157}$$

$$\times \sum_{q=0}^{M} \binom{M}{q} \sum_{n=-\infty}^{\infty} \frac{\Gamma(1+M-n)}{\Gamma(-M-n)} \frac{\Gamma(n-q)n}{\Gamma(n+M-q+1)} t^{n+M-2q} J_n(k\tau) e^{ik\cdot y},$$

a somewhat unwieldy infinite sum over $J_n(k\tau)$. Since the result will be a momentum space two point function, we can express the result of this sum in the following form,

$$v_M = f_M(\omega, k) e^{-i\omega\tau+ik\cdot y}, \tag{158}$$

with the two-point function $f_M$ (up to momentum conservation delta functions) to be determined. We have already computed $f_0$ in the main text. We now proceed inductively. Our result hinges on the following relation,

$$-\Box v_M = \frac{Q_M}{Q_{M+1}} v_{M+1}, \quad \text{where} \quad Q_M \equiv \frac{(-1)^M}{(2M-1)!!(2M+1)!!}, \tag{159}$$

which we will shortly prove using (157). Once this is established, by (158) we have

$$f_{M+1} = \frac{Q_{M+1}}{Q_M} (k^2 - \omega^2) f_M, \tag{160}$$

and with $f_0 = \pm(k^2 - \omega^2)^{\frac{1}{2}}$ we have our final result,

$$f_M = \pm \frac{(-1)^M}{(2M-1)!!(2M+1)!!}\left(k^2 - \omega^2\right)^{\frac{1+2M}{2}} . \tag{161}$$

This matches the known result for a CFT two point function in momentum space (85) up to normalisation.

### D.0.1 Proof of (159)

Let us write (157) as follows,

$$v_M = k^M \tau^{-1-M} \sum_{n=-\infty}^{\infty} a_{Mn} J_n(k\tau) e^{ik\cdot y} , \tag{162}$$

where $a_{Mn}$ are $\tau$-independent coefficients that which can be easily read off from (157),

$$
\begin{aligned}
a_{Mn} &= \left(\frac{1}{2}\right)^M 2^{-1-2M} \frac{\Gamma(1/2-M)}{\Gamma(3/2+M)} \frac{\Gamma(1+M-n)}{\Gamma(-M-n)} \sum_{q=0}^{M} \binom{M}{q} \frac{\Gamma(n-q)n}{\Gamma(n+M-q+1)} t^{n+M-2q} \\
&= \frac{t^{M+n}}{2^{1+3M}} \frac{\Gamma(1/2-M)\Gamma(1+M-n)\Gamma(1+n)}{\Gamma(3/2+M)\Gamma(-M-n)\Gamma(1+n+M)} {}_2F_1\left(-M, -M-n, 1-n, -\frac{1}{t^2}\right) .
\end{aligned}
\tag{163}
$$

The calculation then proceeds by computing the left hand side and right hand side of (159) and showing equality.

**Left hand side.** Compute $\Box v_M = (-k^2 - \partial_\tau^2)v_M$. Where $\tau$ derivatives act on $J_n(k\tau)$ they can be removed by successive use of the formula,

$$J_n'(z) = \frac{1}{2}\left(J_{n-1}(z) - J_{n+1}(z)\right) . \tag{164}$$

What remains is an expression with mixed index $J_n$ with non-homogeneous powers of $\tau$ in the prefactors. Next, homogenise the powers of $\tau$ by raising powers of $\tau$ where appropriate using the identity (154). Doing this step a handful of times gives an expression whose coefficients are all proportional to $\tau^{-1-M}$. Finally, shift the summation index $n$ such that all terms appear as $J_n$. The result is,

$$
\begin{aligned}
\Box v_M = \quad & -k^{2+M}\tau^{-1-M} \sum_{n=-\infty}^{\infty} \left(\frac{(M+n)(M+n-1)}{4(n-1)(n-2)} a_{M,n-2}\right. \\
& \left. + \frac{n^2+M^2+M-1}{2(n^2-1)} a_{M,n} + \frac{(M-n)(M-n-1)}{4(n+1)(n+2)} a_{M,n+2}\right) J_n(k\tau) e^{ik\cdot y} .
\end{aligned}
\tag{165}
$$

**Right hand side.** Starting with $v_{M+1}$ raise the powers of $\tau$ from $\tau^{-2-M}$ to $\tau^{-1-M}$ using (154), and subsequently adjust the summation index $n$ such that all terms once again appear as $J_n$. The result is

$$v_{M+1} = \frac{k^{2+M}\tau^{-1-M}}{2} \sum_{n=-\infty}^{\infty} \left(\frac{a_{M+1,n-1}}{n-1} + \frac{a_{M+1,n+1}}{n+1}\right) J_n(k\tau) e^{ik\cdot y} . \tag{166}$$

**Equality.** By comparing the coefficients of $J_n$ in (165) and in (166) using the expression for $a_{Mn}$ in (163) one can easily verify that (159) holds.

# E   Regularity conditions for normalisable modes

For the normalisable mode computations presented in this paper we have required a regularity condition in the bulk. The purpose of this section is to provide further technical details on this procedure.

We begin with separation of variables, (47). To assess regularity of the resulting mode we turn to the ingoing coordinates presented in Appendix A. This involves changing $z, \eta$ to $x, v$ where $x$ is the new radial coordinate and $v$ labels the ingoing null slice. There are non-regular terms at $x = x_* \equiv 1/3$ which must be removed. The computation is performed order-by-order in $m$. We expand both the radial field $\Phi_{k,\lambda}(z)$ and the Bessel function $\eta J_v(k\eta)$ order-by-order in $m$. To expand the Bessel note that near $x = x_*$ we have $\eta \sim (x - x_*)^{\frac{1}{2}} \left(1 + \sum_k^\infty g_k(x) \left(\frac{m}{H}\right)^k\right)$ where $g(k)$ are finite around $x = x_*$. We also perturbatively correct the index of the Bessel, $v = \sum_k v_k \left(\frac{m}{H}\right)^k$ where $\lambda = H^2(1 - v^2)$. At leading order in $m$ eliminating the non-regular terms at $x = x_*$ – which take the form $(x - x_*)^p$ and $\log(x - x_*)$ – requires that $v_0 \in \mathbb{Z}$. To go to higher orders in $m$ requires that the index of the Bessel is expanded perturbatively around this integer value $v_0$. Thus in doing so we are led to evaluate expressions of the form for small $\epsilon$,

$$J_{v_0+\epsilon}(k\eta) \approx J_{v_0}(k\eta) + \epsilon \partial_v J_v(k\eta)\big|_{v=v_0} + \epsilon^2 \frac{1}{2!} \partial_v^2 J_v(k\eta)\big|_{v=v_0} + O(\epsilon)^3. \tag{167}$$

A compendium of these derivatives can be found in [54]. In summary, analysing the regularity involves studying Bessel functions and their derivatives for small argument.

For the tensors, the procedure to turn off the sources is clear since the $\gamma_{\mu\nu}$ are gauge invariant fluctuations. The solution of the second order differential equation for the radial dependence produces two constants at each order, which together with the $\lambda_k$ correction at the given $k$th order, are the only free parameters. Expanding around the boundary $x = 0$ (47), the sourcelessness condition is imposed by requiring the vanishing of the coefficient going with $x$. This already fixes one constant of integration in terms of the others. The regularity condition gives the remaining constraint, allowing us to determine $\lambda_k$ and leaving a free parameter which gives the amplitude of the mode at a given order.

For the scalars, the sourcelessness and regularity conditions are imposed as follows. We solve the second order equation for the gauge invariant combination $\hat{\phi}$, getting two free parameters at each order in $m$. With it we solve the equation for $S$, getting one extra parameter and we completely determine the gauge invariant combination $\zeta$. Therefore, together with $\lambda_k$, we have four undetermined parameters at each order. We impose regularity around $x_*$ in $\hat{\phi}$. We impose the vanishing of the coefficient going with $x$ in the near boundary expansion of $S$ and the vanishing of the constant coefficient in the near boundary expansion of $\zeta$. These three conditions determine $\lambda_k$ and impose two extra relations on the remaining constants, giving as a result a free parameter for the amplitude at each order.

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
