# Peer review of "Massive holographic QFTs in de Sitter"

_SciPost Physics, doi:SciPost Phys. 12, 182 (2022)_

## Round 1 · Referee Report · Anonymous · 2022-3-3

Strengths
1- Clear and explicit exposition.
2- Discusses QFT on de Sitter at strong coupling.
3- Finds interesting results for the correlation function.
Weaknesses
Very minor (see report)
Report
Using a nice set of coordinates, this paper elucidates the global structure of the geometry sourced by a scalar of dual dimension $\Delta=2$ as first reported on in 1707.01030.
Afterwards, one- and two-point functions of scalars and tensor modes in this backgrounds are computed in order to extract strong coupling correlation functions of the stress tensor and dual operator of the scalar for the QFT on dS$_3$, as a function of the mass deformation.
The authors note that the resummation of the order-by-order result can be understood as a shift in the spectral poles of the two-point function, without any additional structure being introduced. The authors also compare these results to a massive fermionic QFT on dS$_3$.
I find these results intriguing and this paper easily meets the criterion of publication in Scipost Physics.
The only request I have is that the authors comment on the strong coupling interpretation of the order by order resummation in $m$ of their two-point function result giving a sum of shifted poles. In the introduction, the authors make clear that they are aware of the IR issues that arise in perturbative calculation of correlation functions on dS at weak coupling. However, they do not touch on this in the resummation section. It would naively seem to me that the shifted poles, particularly if they lie at real values of $\nu$ imply that the strong coupling result is consistent with these field theories being stable.
Requested changes
Comments on the perturbative IR instability in dS in the resummation section.

---

## Round 1 · Referee Report · Anonymous · 2022-3-21

Strengths
1- Clear Presentation of Analytic Computations.
2- Well defined problem and results on the correlation functions.
Report
The authors discuss the strongly coupled holographic computation of the QFT correlation functions on de Sitter spacetime. To capture non-trivial de Sitter effects in their computation, a mass deformation is introduced to the CFT to break the conformal symmetry. In this way an extra scale m is introduced to represent the mass on top of the de Sitter Hubble constant that already exist in the undeformed theory. All the results of the authors are expected to be dependent on the ratio m/H, which can be used as the parameter for perturbative expansions.
The authors present the holographic background by solving perturbatively the gravity equation resulting from a gravity action coupled to a scalar with a mass term. Similar derivations/results already exist in the literature which the authors cite. The initial solution is found perturbatively in coordinates that the boundary covers the de Sitter inflationary patch. The global structure of the spacetime is also presented perturbatively with respect to the mass deformation parameter.
Then the one-point function is obtained on the holographic background. For low values of m/H, it is in agreement with the results of the literature, e.g. ref. [27]. At large values of m/H the asymptotic behavior of the one point function is derived, up to a constant that is determined numerically. The authors proceed to compute the scalar, vector, tensor, two point functions by using the standard holographic methods. That is by solving the equations of motion for the linearised fluctuations on the domain wall background they are working on. The form of the correlators is not fixed in their theories since the conformal symmetry is broken. The perturbative result of the two point function is obtained, this is one of the main results of the paper. The m=0 limit reproduces the CFT results as it should. Then the authors discuss a massive free fermion theory on se Sitter and compare with their holographic results.
The computations of the paper are presented analytically and in a clear way. The problem that the manuscript discusses is well posed and defined and it is answered clearly following the standard methods of holography.
I have some minor suggestions and comments. The authors find that the correlation functions admit a spectral representation as a sum of certain poles corresponding to the normalisable modes of the bulk spacetime. They discuss (the absence of) the interpretation of these modes as quasinormal modes in their computation, mentioning briefly the literature. This discussion can be improved/extended. The thermal behavior of de Sitter is associated with the cosmological horizon seen by the observer. Its characteristic temperature is proportional to the de Sitter radius and the cosmological horizon is observer dependent (unlike the black holes thermal properties). Moreover, the domain walls are de Sitter invariant. Are all these facts essential/enough for the conclusions of the authors in the relevant discussion in section 7, regarding the mentioned unrelatedness to the quasinormal modes. I would suggest a slight extension of the relevant discussion in section 7, and a further elaboration on the interpretation of these results in comparison to the results of zero momenta e.g. ref [27].
The authors could also mention the reasons they have chosen to work in three-dimensional model and if there are obstructions of extending their computation in higher dimensions where cosmological observations are relevant. As an extra minor remark, I find the title of the manuscript to be broader than its contents.
In summary the paper is well written. The question that the manuscript studies is well defined and it is answered clearly. I would suggest a minor revision of the manuscript.
Requested changes
1- Comments on the correlation function representation as a sum of poles, the normalisable modes of the bulk spacetime and the quasinormal modes(see report).

---

## Round 2 · Referee Report · Anonymous · 2022-4-13

Strengths
1- Clear and explicit exposition.
2- Discusses QFT on de Sitter at strong coupling.
3- Finds interesting results for the correlation function.
Report
This paper meets all the criteria for publication in SciPost.
Requested changes
None

---

## Round 2 · Referee Report · Anonymous · 2022-5-4

Strengths
1- Clear Presentation of Analytic Computations.
2- Well defined problem and results on the correlation functions.
Report
The authors have clarified the points raised. This version of the manuscript is suitable for publication in SciPost.

---

## Round 2 · Author Response

We thank the referee for their comments and for their suggestion. We agree that the details of the two-point function with m corrections shed further light on the stability issue. We have added some clarifying comments in section 7 paragraph 4. In particular, the strong coupling result at m=0 is stable and controlled by a CFT. The m corrections are indeed compatible with stability. This can be seen by e.g. looking at the Bessel functions which decay with a power law in conformal time governed by the index. Thus all the poles we have in the paper give rise to decaying modes and no instabilities are observed.
Response to Anonymous Report 2
We thank the referee for their comments and suggestions. Regarding the relation to hydrodynamics we have added some clarifying remarks to section 7 paragraph 5 and added a new paragraph 6. The connection to the temperature seen by a static patch observer is interesting, though this is not an avenue we have explored in this work, where we have mostly been concerned with one- and two-point functions computed in the inflationary patch. It would be interesting to understand the normalisable modes from the static patch perspective and the relation to the cosmological horizon temperature in future work. Finally, we chose to work in three dimensions because the analysis is the simplest in this case and the construction of [27] was already available. We are not aware of any obstructions to carrying out the analogous construction in higher dimensions. We are pursuing this for future work where indeed we hope to make contact with cosmological observations.

---

## Round 2 · List of Changes

- Added clarifying comments surrounding m corrections to pole locations and the stability of the theory, section 7 paragraph 4.
- Added further comments and additional discussion regarding the lack of a hydrodynamic interpretation. Section 7 paragraph 5 and added a new paragraph 6.

You are currently on this page

---

## Editorial Decision

published